# BysR, a LysR-Type Pleiotropic Regulator, Controls Production of Occidiofungin by Activating the LuxR-Type Transcriptional Regulator AmbR1 in *Burkholderia* sp. Strain JP2-270

Lijuan Wu,[a] Liqun Tang,[a] Yuchang He,[a] Cong Han,[a] Lei Wang,[a] Yunzeng Zhang,[b] Zhiguo E[a]

[a]State Key Laboratory of Rice Biology and Breeding, China National Rice Research Institute, Hangzhou, China
[b]Joint International Research Laboratory of Agriculture and Agri-Product Safety, the Ministry of Education of China, Yangzhou University, Yangzhou, China

**ABSTRACT** Occidiofungin is a highly effective antifungal glycopeptide produced by certain *Burkholderia* strains. The *ocf* gene cluster, responsible for occidiofungin biosynthesis, is regulated by the cluster-specific regulators encoded by an *ambR* homolog(s) within the same gene cluster, while the extent to which occidiofungin biosynthesis is connected with the core regulation network remains unknown. Here, we report that the LysR-type regulator BysR acts as a pleiotropic regulator and is essential for occidiofungin biosynthesis. *Magnaporthe oryzae* was used as an antifungal target in this study, and deletion of *bysR* and *ocfE* abolished the antagonistic activity against *M. oryzae* in *Burkholderia* sp. strain JP2-270. The Δ*bysR* defect can be recovered by constitutively expressing *bysR* or *ambR1*, but not *ambR2*. Electrophoretic mobility shift assays (EMSAs) collectively showed that BysR regulates *ambR1* by directly binding to its promoter region. In addition, transcriptomic analysis revealed altered expression of 350 genes in response to *bysR* deletion, and the genes engaged in flagellar assembly and bacterial chemotaxis constitute the most enriched pathways. Also, 400 putative BysR-targeted loci were identified by DNA affinity purification sequencing (DAP-seq) in JP2-270. These loci include not only genes engaged in key metabolic pathways but also those involved in secondary metabolic pathways. To conclude, the occidiofungin produced by JP2-270 is the main substance inhibiting *M. oryzae*, and BysR controls occidiofungin production by directly targeting *ambR1*, an intracluster transcriptional regulatory gene that further activates the transcription of the *ocf* gene cluster.

**IMPORTANCE** We report for the first time that occidiofungin production is regulated by the global transcriptional factor BysR, by directly targeting the specific regulator *ambR1*, which further promotes the transcription of *ocf* genes. BysR also acts as a pleiotropic regulator that controls various cellular processes in *Burkholderia* sp. strain JP2-270. This study provides insight into the regulatory mechanism of occidiofungin synthesis and enhances our understanding of the regulatory patterns of the LysR-type regulator.

**KEYWORDS** occidiofungin, LysR-type regulator, AmbR1, pleiotropic regulator, inhibitory activity

Address correspondence to Zhiguo E, ezhiguo@caas.cn, or Yunzeng Zhang, yzzhang@yzu.edu.cn.

The authors declare no conflict of interest.

The pathogenic fungus *Magnaporthe oryzae* is the causal agent of rice blast disease, which causes severe losses in grain yield annually. To manage blast disease, application of biocontrol strains of *Burkholderia*, which belongs to the betaproteobacteria, is a powerful strategy in addition to chemical fungicides and plant resistance breeding (1, 2). The *Burkholderia cepacia* complex (Bcc) represents a diverse group of ubiquitously distributed bacteria in various niches, such as soil, water, plants, animals and humans (3). The Bcc has significant biotechnological potential as a source of novel antibiotics and bioactive secondary metabolites. Multiple bioactive secondary metabolites,

such as pyrrolnitrin, occidiofungin, burkholdines, gladiolin, enacyloxin IIa, and cepacins, which have shown great potential in biocontrol and pharmaceuticals, have been identified in organisms affiliated with the Bcc (4–7).

Occidiofungin, a glycopeptide, was first identified from *Burkholderia contaminans* MS14 (8), which possesses significant antifungal and antiparasitic activities (2, 8, 9). Occidiofungin is biosynthesized by *ocfD*, *ocfE*, *ocfF*, *ocfH*, and *ocfJ*, which encode nonribosomal peptide synthetases (10, 11). Previous comparative genomics analysis suggested that the occidiofungin biosynthesis genes *ocfD* to *ocfJ* constitute a gene cluster and are present in plant growth-promoting *Burkholderia* strains but absent in pathogenic strains or non-plant-associated soil strains (12). The expression of *ocfD* to *ocfJ* is suggested to be specifically regulated by two LuxR-type regulatory genes, *ambR1* and *ambR2*, located in the gene cluster with *ocfD* to *ocfJ*. AmbR1 plays a more critical role than AmbR2 (10, 11, 13). However, the integration mechanism of this secondary metabolism pathway with the core regulation network remains unknown.

In the elite biocontrol strain *Burkholderia* sp. strain JP2-270, we recently identified a transcriptional regulator, BysR, that is essential for the fungal-inhibitory activity of JP2-270 (14). BysR belongs to the LysR-type transcriptional regulators (LTTRs). LTTRs are the most common kind of transcription regulators in prokaryotes and can act as positive or negative modulators for various genes involved in symbiotic nitrogen fixation, virulence, quorum sensing, motility, and secondary metabolism (15–23). The LTTR StgR inhibits the production of actinorhodin and prodigiosin by interacting with their pathway-specific regulators in *Streptomyces coelicolor* (23), while the LTTR FinR was recently reported to serve as an activator of phenazine and pyoluteorin biosynthesis in *Pseudomonas chlororaphis* G05 (22). In addition, ScmR was identified as a global LTTR controlling virulence and diverse secondary metabolites in *Burkholderia thailandensis* (17).

Our work aimed to investigate the novel LTTR BysR, which is essential for the antifungal activity of *Burkholderia* sp. strain JP2-270. By inactivating *ocfE*, we demonstrated that occidiofungin produced by JP2-270 mainly contributes to the suppression activity against *M. oryzae*. In addition, we found that BysR is required to inhibit the mycelial growth of *M. oryzae*. To understand the functions and regulatory mechanisms of BysR in occidiofungin production, as well as other pathways, we performed electrophoretic mobility shift assays (EMSAs), RNA sequencing (RNA-seq), and DNA affinity purification sequencing (DAP-seq). The EMSAs showed that occidiofungin synthesis was regulated by BysR, which directly targeted *ambR1*, a pathway-specific regulatory gene of occidiofungin production. Aside from secondary metabolism, a series of regulatory targets were identified by DAP-seq, including genes engaged in core cellular processes and secondary metabolites. In general, our results identified a novel upstream component of the occidiofungin synthesis regulatory network and provided new insight into the regulatory mechanisms of occidiofungin synthesis. The regulatory models have been proposed to help us understand the global regulatory role of the LTTR BysR in diverse cellular processes.

## RESULTS

**BysR displays conserved domains of LTTR regulators.** *bysR* (DM992_17470) has an open reading frame of 984 bp and encodes a putative 327-amino-acid protein. The phylogenetic analysis of BysR with its most relevant LTTRs revealed that BysR is a novel LTTR in the genus *Burkholderia*, and the protein that is most homologous to BysR is Bcal3178 of *Burkholderia cenocepacia* H111, showing 93% identity (96% similarity) (see Fig. S1A in the supplemental material). The LTTRs associated with secondary metabolite synthesis are ScmR of *Burkholderia thailandensis* and ShvR of *Burkholderia cenocepacia* (17, 24) (Fig. S1A). A domain homology search with Pfam (25) revealed a typical helix-turn-helix (HTH) DNA binding domain (Pfam accession number PF00126) at the N terminus and a typical C-terminal receiver domain (PF03466) (Fig. S1B). In addition, the homology comparison at the amino acid level between BysR and other identified LTTRs revealed that the high degree of similarity was within the N-terminal helix-turn-

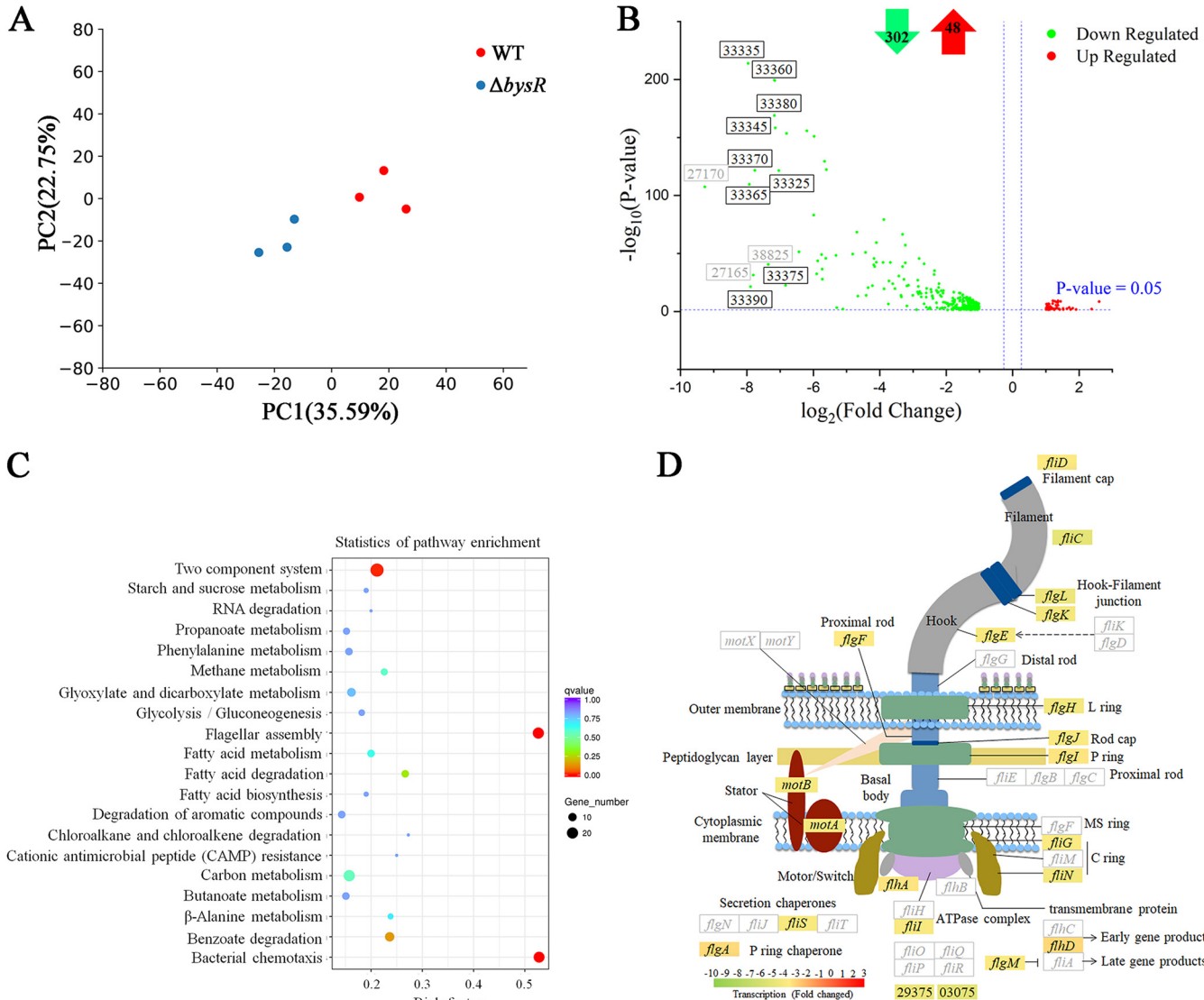

**FIG 1** Identification of BysR-dependent genes in *Burkholderia* sp. strain JP2-270. (A) PCA of six samples from RNA-seq analysis. The blue and red dots represent three different replicates of Δ*bysR* and WT treatment groups, respectively. (B) Volcano plot of RNA-seq data comparing the transcriptomes of the Δ*bysR* mutant and the WT. Differentially expressed genes [|log$_2$(fold change)| > 1, and FDR-adjusted $P \leq 0.05$] are indicated with green dots (for downregulated genes) and red dots (for upregulated genes). The genes with the most significant downregulation of expression are labeled. The numeric string includes the last 5 digits of the locus tag of the gene in NCBI; e.g., 33335 represent DM992_33335. Gene names in black indicate genes involved in occidiofungin synthesis; gene names in gray indicate genes associated with other biological processes. The numbers of up- and downregulated genes are shown at the top in red and green arrows, respectively. (C) Statistical enrichment of differentially expressed genes in KEGG pathways. The 20 most significantly enriched pathway items are shown. The rich factor represents the ratio of the number of differentially expressed genes enriched in this pathway to the number of background genes. (D) The flagellar assembly model and the expression of related genes. Colored rectangles represent genes regulated by BysR based on the RNA-seq results, and the color scale indicates the relative transcriptional level [log$_2$(fold change)] based on RNA-seq results. The genes involved in flagellar assembly are labeled.

helix domain involved in binding DNA and the less conserved region was in the C terminus, where the coinducer domain was located (Fig. S1C).

**BysR positively regulates multiple pathways in *Burkholderia* sp. strain JP2-270.** Here, we performed RNA-seq transcriptome analysis to identify the potential targets regulated by BysR. The good grouping in principal-component analysis (PCA) and high Pearson correlation coefficient indicated that the data had good discriminability and reproducibility (Fig. 1A and Fig. S2). Furthermore, the expression of some genes was similar in RNA-seq and quantitative reverse transcription-PCR (qRT-PCR) (Fig. S3), suggesting that the data were reliable. RNA-seq analysis revealed 350 differentially expressed genes (DEGs), consisting of 48 upregulated and 302 downregulated genes [with adjusted $P$ ($P_{adj}$)

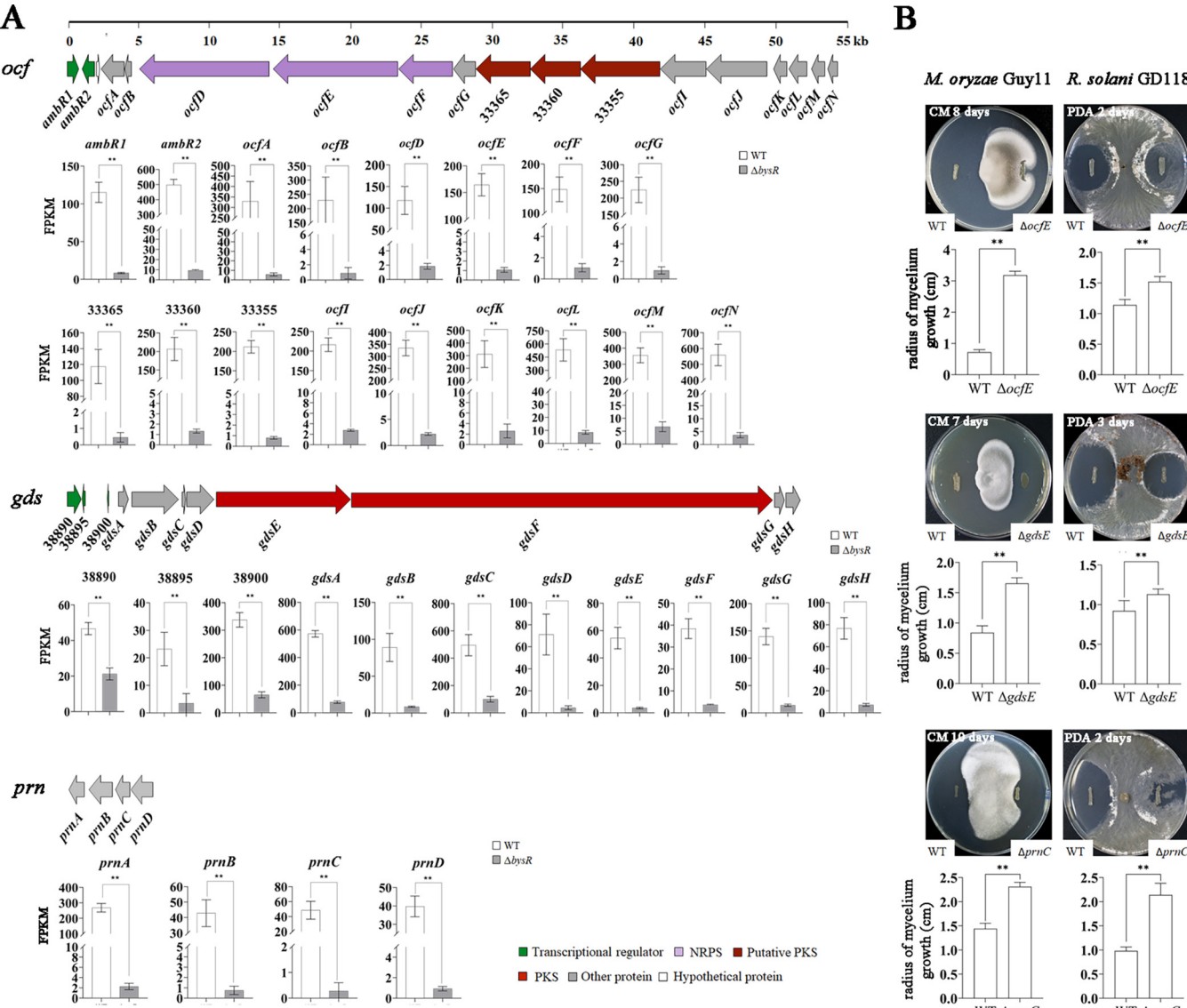

**FIG 2** Gene clusters of secondary metabolite synthesis regulated by BysR. (A) Organization of occidiofungin (*ocf*), gladiostatin (*gds*), and pyrrolnitrin (*prn*) genes and expression levels (FPKM) of genes in *ocf*, *gds*, and *prn* gene clusters based on RNA-seq. (B) Antifungal activity analysis of key gene deletion mutants in *ocf*, *gds*, and *prn* gene clusters. WT, wild type; Δ*ocfE*, the mutant with an in-frame deletion of *ocfE*, encoding NRPS; Δ*gdsE*, mutant with an in-frame deletion of *gdsE*, encoding PKS; Δ*prnC*, mutant with an in-frame deletion of *prnC*, encoding FAD-dependent oxidoreductase.

values of <0.05 and |log₂(fold change)| values of (>1) (Fig. 1B; Table S2)]. Among the 12 most significant downregulated genes, 9 were related to occidiofungin production (DM992_33325, DM992_33335, DM992_33345, DM992_33360 to DM992_33380, and DM992_33390). As shown in Fig. 2A, the FPKM (fragments per kilobase of transcript sequence per million base pairs sequenced) values of genes related to occidiofungin synthesis in a Δ*bysR* mutant were significantly lower than those in wild-type JP2-270, and these genes were dramatically downregulated 3.6- to 7.9-fold in the Δ*bysR* mutant (Fig. 1B). Among them, besides the dramatically affected occidiofungin production-associated gene cluster, four other secondary metabolite biosynthesis gene clusters, including an unknown nonribosomal peptide synthetase (NRPS) type gene cluster (DM992_33030 to DM992_33110, region 3.1 [Table 1]), an unknown bacteriocin type gene cluster (DM992_26140 to DM992_26160, region 2.6 [Table 1]), a trans-acyltransferase polyketide synthase (trans-AT PKS) gene cluster (DM992_38905 to DM992_38940, region 3.7 [Table 1]) involved in gladiostatin production, and a pyrrolnitrin biosynthesis gene cluster (DM992_38820 to DM992_38835, region 3.6 [Table 1]) were also

**TABLE 1** Gene clusters of *Burkholderia* sp. strain JP2-270 predicted by antiSMASH and BLAST analysis[a]

| Cluster | Type | Position | Most similar known cluster | Type of compounds | % similarity |
|---|---|---|---|---|---|
| Chr1 (CP029824) | | | | | |
| Region 1.1 | T1PKS | 1–32,032 | Capsular polysaccharide | Saccharide | 16 |
| Region 1.2 | Arylpolyene | 1,249,332–1,290,543 | APE Vf | Other | 10 |
| Region 1.3 | Terpene | 2,271,135–2,291,968 | | | |
| Region 1.4 | NRPS | 2,482,384–2,529,203 | Burkholderic acid | NRP + polyketide:modular type I | 31 |
| Region 1.5 | NRPS | 2,723,377–2,778,049 | Ornibactin | NRP | 100 |
| Chr2 (CP029825) | | | | | |
| Region 2.1 | Terpene | 275,900–296,964 | *N*-Acyloxyacyl glutamine | Other | 50 |
| Region 2.2 | Phosphonate | 418,602–460,290 | Phosphinothricintripeptide | NRP | 6 |
| Region 2.3 | Redox cofactor | 1,186,903–1,209,041 | | | |
| Region 2.4 | NRPS-like | 1,651,742–1,709,599 | Malleobactin A to D | NRP:NRP siderophore | 22 |
| Region 2.5 | Terpene | 1,784,391–1,808,491 | | | |
| Region 2.6 | RRE-containing, RiPP-like | 1,834,070–1,847,121 | | | |
| Region 2.7 | Phenazine | 1,874,156–1,894,584 | | | |
| Region 2.8 | Homoserine lactone | 2,804,503–2,825,111 | | | |
| Region 2.9 | Ectoine | 3,028,116–3,038,514 | | | |
| Chr3 (CP029826) | | | | | |
| Region 3.1 | TransAT-PKS, NRPS, NRPS-like | 1–47,903 | Lagriamide | Polyketide | 9 |
| Region 3.2 | NRPS-like, T1PKS, NRPS | 77,377–163,367 | Occidiofungin A | NRP + polyketide | 88 |
| Region 3.3 | Terpene | 448,407–470,461 | | | |
| Region 3.4 | Homoserine lactone | 707,640–728,338 | | | |
| Region 3.5 | RiPP-like | 1,069,004–1,079,819 | | | |
| Region 3.6 | Other | 1,328,932–1,360,585 | Pyrrolnitrin | Other | 100 |
| Region 3.7 | TransAT-PKS | 1,360,644–1,429,994 | Gladiostatin | Polyketide | 100 |
| Plas2 (CP029828) | | | | | |
| Region 5.1 | Other | 63,674–106,502 | | | |
| Region 5.2 | Other | 166,130–208,973 | O antigen | Saccharide | 10 |

[a]Chr, chromosome; T1PKS, Type I Polyketide synthase; TransAT-PKS, trans-acyltransferase polyketide synthase; APE, Aryl polyene; RRE, RiPP recognition element; RiPP, Ribosomally synthesized and post-translationally modified peptide.

identified (Fig. 2A and Table S2). These results suggest that BysR regulates the production of secondary metabolites in *Burkholderia* sp. strain JP2-270. Otherwise, among the upregulated genes, those responsible for the biosynthesis of pyrroloquinoline quinone (PQQ) were affected significantly (Table S2), suggesting that BysR might act as a negative regulator of PQQ production.

Moreover, KEGG pathway enrichment analysis of DEGs revealed that a variety of cellular processes involved in carbon metabolism, chloroalkane and chloroalkene degradation, two-component systems, chemotaxis, and flagellar assembly were significantly enriched (Fig. 1C and Table S3). Notably, 20 of the 38 genes engaged in flagellar assembly were downregulated in the Δ*bysR* mutant (Fig. 1D; Tables S2 and S3), indicating that BysR plays an important role in regulating flagellar assembly.

The DEGs identified by RNA-seq include genes that are regulated directly and indirectly by BysR. To verify the genes that are directly regulated by BysR, we used DNA affinity purification sequencing analysis (DAP-seq) for genome-wide recognition of BysR binding sites *in vitro* (26). After affinity purification and sequencing, at least 22 million double-end reads per sample were generated, and the ones with >99% of reads were uniquely mapped to the JP2-270 genome. A total of 400 enriched common peaks of two replicates with a $-\log_{10}(P$ value) of $\geq 2$ were called (Table S4). The mean width of DAP-seq peaks was <1,000 bp (Fig. 3A). In total, 89.6% of BysR binding peaks were distributed in the promoter, promoter-transcription start site (TSS), or intergenic regions, while only 8.3% and 2.3% were in the transcription termination site (TTS) and exon regions, respectively (Fig. 3B). Among the identified peaks, 232 (58%) were located in the −700 bp-to-100 bp regions, as determined by analysis of peak summit positions relative to the start codons of JP2-270 open reading frames (or the first gene in operons) (Fig. 3C; Table S4). These results suggest that the expression of these loci might be directly regulated by BysR. Among the 232 peaks, 134

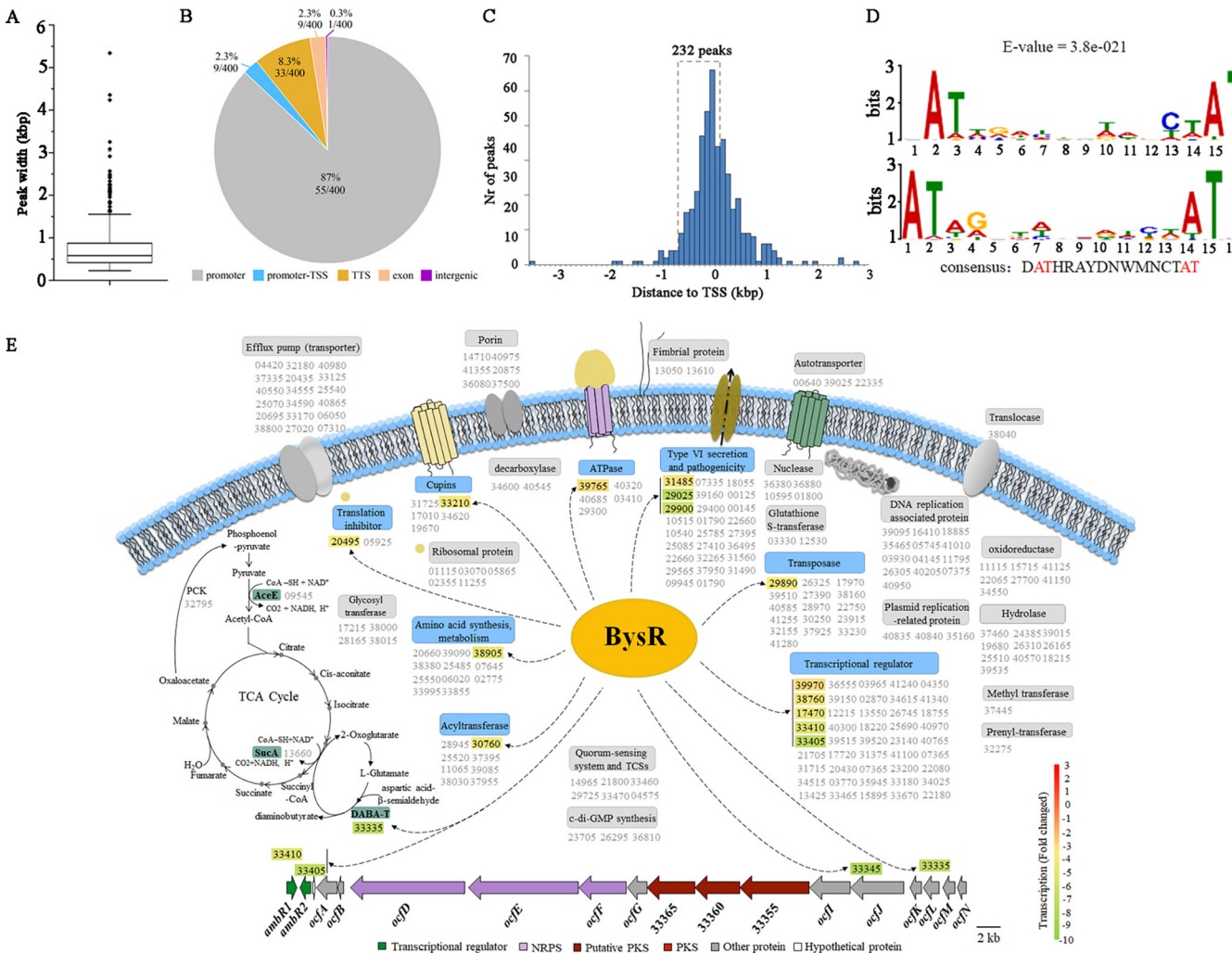

**FIG 3** Identification of genes directly regulated by BysR in *Burkholderia* sp. strain JP2-270. (A) Widths of BysR DAP-seq peaks. (B) Number and distribution of BysR binding genes obtained by DAP-seq. (C) Distribution of the distance between DAP-seq peak summit positions relative to the start codons of JP2-270 open reading frames. Nr, number. (D) Consensus binding sequence of BysR. The reverse complement of this sequence is presented at the bottom. (E) Proposed model for the regulation of various cellular processes by BysR in *Burkholderia* sp. strain JP2-270. The black dotted arrows indicates directly activation by BysR. The numeric strings are the last 5 digits of the locus tag of the gene in NCBI; e.g., 33335 represents DM992_33335. Gene names in gray indicate genes putatively directly regulated by BysR based only on DAP-seq results. Gene names in black on a colored background represent genes putatively directly regulated by BysR based on both the RNA-seq and DAP-seq results. The color scale indicates the relative transcriptional level [log₂(fold change)] of the genes in the Δ*bysR* mutant relative to the WT based on RNA-seq results.

peaks with a length greater than 500 bp were used for motif prediction using MEME-ChIP (27) and the conserved-motif-like AT-N$_{11}$-AT box (E value = 3.8−021) was found in 95 of 134 peaks (71%) (Fig. 3D and Table S5). This conserved A+T-rich box was similar to the LTTR OxyR consensus binding sequence (28), and the typical motif of LTTRs (T-N$_{11}$-A) was also recognized in this conserved box (18). Thus, we propose that the AT-N$_{11}$-AT box is the binding feature of BysR (Fig. 3D). Further, the potential BysR binding sequences in 95 peaks were searched by using the regular expression module in Python, and at least one AT-N$_{11}$-AT box sequence was found in each of these 95 peaks (Table S5), implying that BysR binds around the proposed consensus loci to regulate the transcription of the target genes. In order to confirm that the AT-N$_{11}$-AT box sequences are essential for BysR binding, the promoter region fragments of genes DM992_38905 and DM992_31485, with wild-type and mutated sequences of motif-like consensus sequences, were labeled with Cy5 and used for EMSA analysis. The presence of shift bands in the wild-type probes (probe 4 and probe 6) and the absence of BysR binding to the mutated probes (probe 5 and probe 7) indicated that the predicted consensus-binding sequences were necessary for BysR to bind to the promoter region and regulate the transcription of targeted genes

(Fig. S4). In addition, the identification of the *ambR1* promoter locus (peak ID: Merged-Chr3-148813-2 [Table S4; Fig. 3E]) indicated that BysR may regulate the expression of the *ocf* gene cluster by directly binding to the *ambR1* promoter region.

Our data also suggested that BysR may direct regulate genes involved in core metabolisms, such as the tricarboxylic acid (TCA) cycle, amino acid metabolism, and DNA replication (Fig. 3E and Table S4). Similarly, the growth rate of the Δ*bysR* mutant was slightly lower that of wild-type JP2-270, and the biomass of the Δ*bysR* mutant was less than that of JP2-270 in a stable period when cultured in rich medium (Fig. S5), indicating that BysR affect the core metabolism of JP2-270. In addition, some genes related to drug resistance, transport, mobility, transposase activity, transcriptional regulation, and quorum sensing were also identified as the direct targets of BysR (Fig. 3E and Table S4).

Overall, the results suggest that BysR might be a pleiotropic transcriptional factor that directly and indirectly regulates gene expression in a variety of cellular processes in *Burkholderia* sp. strain JP2-270.

**BysR plays an important role in the broad-spectrum antifungal activity of JP2-270.** In our previous research, we showed that JP2-270 could suppress the growth of *Rhizoctonia solani* GD118 (14). Furthermore, in this study, we also assessed the inhibitory activity of JP2-270 against *M. oryzae* Guy11. As shown in Fig. 4A and J, the mycelial growth of Guy11 was obviously inhibited by JP2-270, with the mycelial growth radius of Guy11 being $0.72 \pm 0.08$ cm and $3.44 \pm 0.10$ cm after 7 days of incubation with and without JP2-270, respectively ($P < 0.01$) (Fig. 4A, B, and J). The strong antagonistic activity against *M. oryzae* and *R. solani* indicated that JP2-270 had a wide-spectrum antifungal activity.

As a pleiotropic transcriptional factor, BysR could regulate various cellular processes, including secondary metabolism. In this study, we also showed that the Δ*bysR* mutant almost totally lost inhibitory activity against *M. oryzae* (Fig. 4C and J). The reduced suppressive activity of the Δ*bysR* strain could be restored by introducing pBBR2-*bysR* expressing wild-type *bysR* but not by introducing the empty vector pBBR1MCS-2 (29) (Fig. 4D, E, and J). These results further confirmed that *bysR* is necessary for antifungal activity of JP2-270.

**Occidiofungin is the main secondary metabolite with antifungal activities.** As mentioned above, we found that the expression levels (based on FPKM value) of several genes involved in occidiofungin (region 3.2), gladiostatin (region 3.7), and pyrrolnitrin (region 3.6) biosynthesis were dramatically decreased in the Δ*bysR* mutant (Fig. 2A; Table S2). To understand the role of these secondary metabolites in inhibiting the mycelial growth of fungi, markerless mutants with deletion of *ocfE*, encoding nonribosomal peptide synthetase, *gdsE*, encoding polyketide synthase, and *prnC*, encoding FAD-dependent oxidoreductase, were constructed, and the inhibitory activities of derivates were also tested. Based on the inhibition assays, we observed that the mycelium growth radius of Guy11 was $3.18 \pm 0.13$ cm for the Δ*ocfE* strain (compared to $0.72 \pm 0.08$ cm for JP2-270), $1.65 \pm 0.09$ cm for the Δ*gdsE* strain (compared to $0.84 \pm 0.11$ cm for JP2-270), and $2.31 \pm 0.09$ cm for the Δ*prnC* strain (compared to $1.44 \pm 0.11$ cm for JP2-270) (Fig. 2B). The mutant with *ocfE* deletion exhibited weaker antifungal activity against *M. oryzae* than the Δ*gdsE* and Δ*prnC* mutants, indicating that occidiofungin produced by JP2-270 plays a more important role than gladiostatin and pyrrolnitrin in inhibiting mycelial growth of *M. oryzae*. Pyrrolnitrin produced by JP2-270 was the main metabolite that inhibited *R. solani*, with mycelium growth radii of $0.98 \pm 0.08$ cm and $2.14 \pm 0.24$ cm for JP2-270 and the Δ*prnC* mutant, respectively. Compared to pyrrolnitrin, occidiofungin and gladiostatin showed lower activity against *R. solani*, with radii of $1.52 \pm 0.08$ cm for the Δ*ocfE* mutant (compared to $1.14 \pm 0.09$ cm for JP2-270) and $1.13 \pm 0.07$ cm for the Δ*gdsE* mutant (compared to $0.92 \pm 0.13$ cm for JP2-270). Overall, occidiofungin and pyrrolnitrin had better inhibitory activity against *M. oryzae* and *R. solani*, respectively, while gladiostatin showed moderate inhibitory activity against these fungi (Fig. 2B). Collectively, these results showed that JP2-270 could produce a variety of secondary metabolites, among which occidiofungin is critical for inhibiting *M. oryzae*.

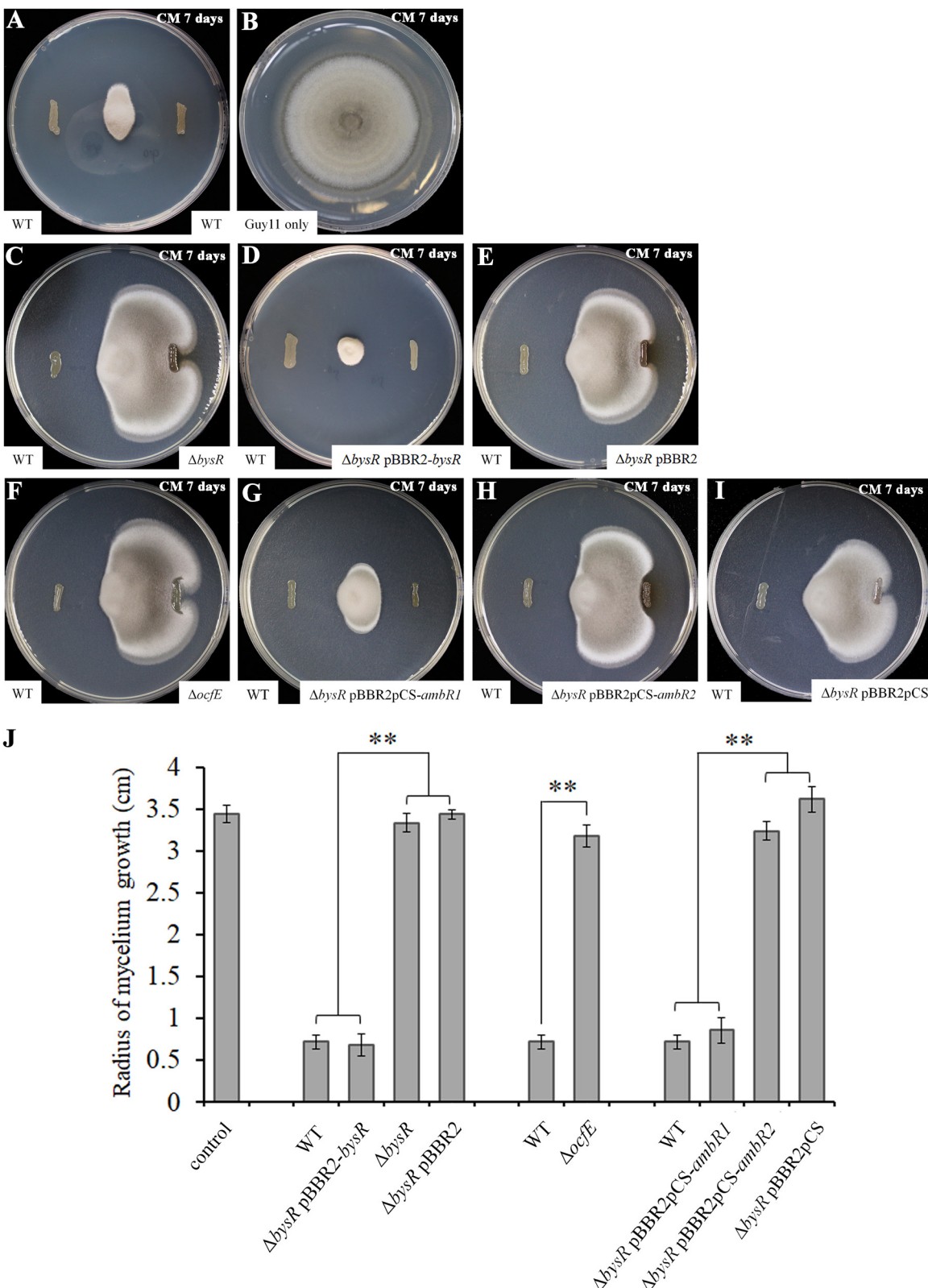

**FIG 4** Inhibitory activity of *Burkholderia* sp. strain JP2-270 and its derivates against *M. oryzae*. (A) Mycelial growth of *M. oryzae* was inhibited by wild-type isolate JP2-270 (WT). (B) Uninhibited mycelial growth of *M. oryzae* was used as the control. (C) Mycelial growth of *M. oryzae* was inhibited by WT JP2-270 and the Δ*bysR* mutant. (D) Mycelial growth of *M. oryzae* was inhibited by WT JP2-270 and the Δ*bysR* pBBR2-*bysR* mutant. (E) Mycelial growth of *M. oryzae* was inhibited by WT JP2-270 and the Δ*bysR* pBBR2 mutant. (F) Mycelial growth of *M. oryzae* was inhibited by WT JP2-270 and the Δ*ocfE* mutant. (G) Mycelial growth of *M. oryzae* was inhibited by

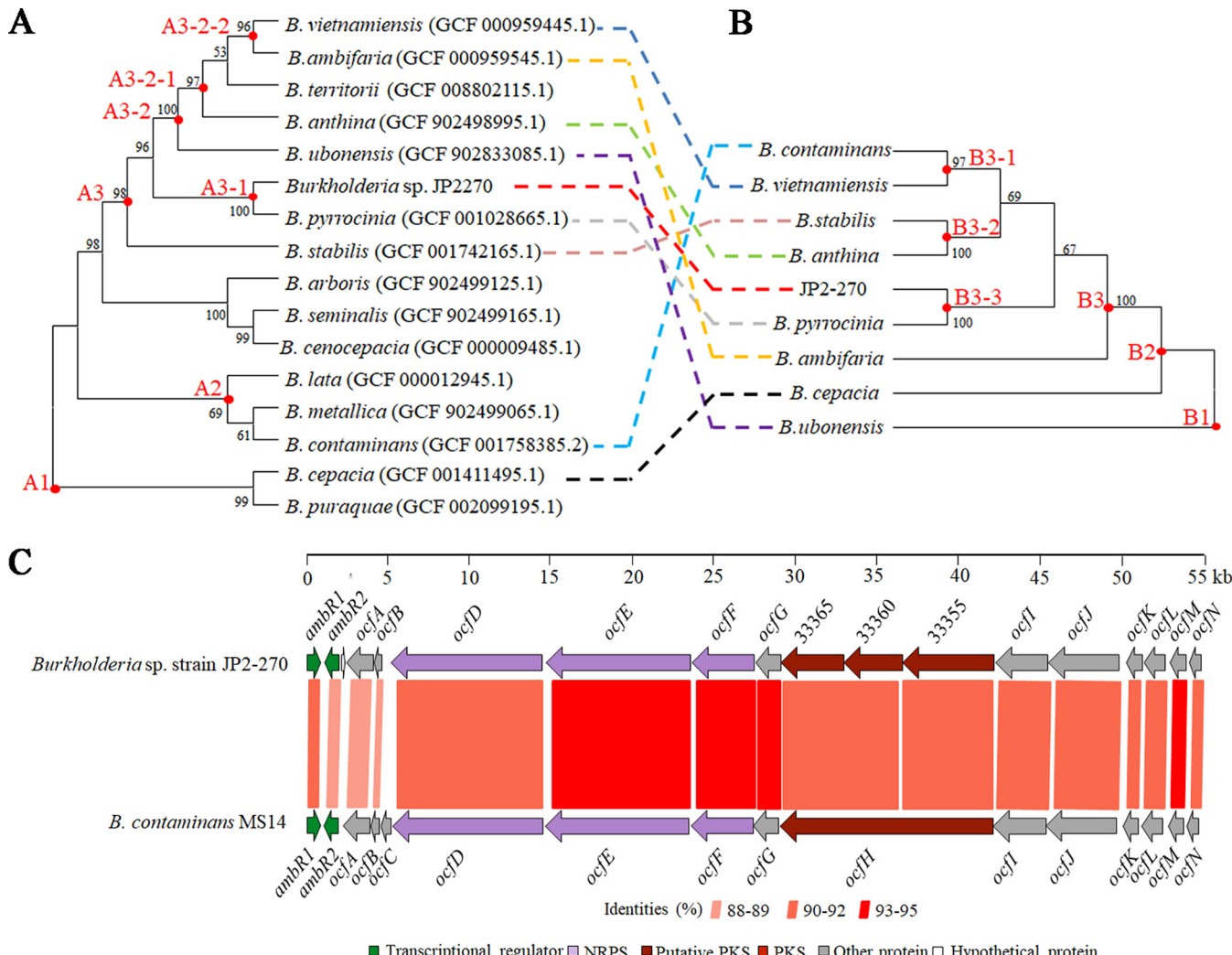

**FIG 5** Comparative evolutionary analysis based on genome-wide and Ocf proteins. (A) Phylogenetic tree of genome-wide protein-coding genes. The phylogenetic relationship was inferred by MEGA7 using maximum parsimony (MP) with the tree bisection regrafting (TBR) algorithm (56). The most parsimonious tree with a length of 45,891 is shown. PhyloPhlAn 3 was used to select conserved marker genes encoding a total of 20,092 amino acids in the genome sequences of representative species belonging to *Burkholderia*. The combined sequences of all marker genes were used for phylogenetic analysis. The numbers above branches are bootstrap support values of >60% from 1,000 replications, with an average branch support of 90%. (B) Phylogenetic tree of Ocf proteins generated by MEGA7 (56). The Ocf proteins included the concatenated sequences of AmbR1/2 and OcfA to OcfN. Ocf proteins were found only in these 8 known species of *Burkholderia* and JP2-270. The amino acid substitution model was the Poisson model. The evolutionary history was inferred using MP with the TBR algorithm. The most parsimonious tree with a length of 10,535 is shown. Bootstrap values (>60%) from 1,000 replicates are shown at the nodes. (C) Comparison of the gene organization for the *ocf* gene clusters of *Burkholderia* sp. strain JP2-270 and *B. contaminans* MS14. The predicted occidiofungin biosynthesis genes were highly conserved between JP2-270 and MS14.

**Comparative evolutionary analysis based on genome and Ocf proteins.** Phylograms inferred from genome sequences revealed that the species closest to JP2-270 is *Burkholderia pyrrocinia*, and both belong to the Bcc (Fig. 5A). Occidiofungin was originally isolated from *B. contaminans* MS14, and we showed that the *ocf* gene cluster also occurred in 8 other species of *Burkholderia* (Fig. 5B). The evolutionary trajectory of Ocf proteins from some species was inconsistent with the genome-wide evolution of the corresponding host strains. *B. contaminans* and *Burkholderia vietnamiensis* were clustered in subclade B3-1 (Fig. 5B), but they were distributed separately in subclades A2 and

**FIG 4 Legend (Continued)**
WT JP2-270 and the Δ*bysR* pBBR2pCS-*ambR1* mutant. (H) Mycelial growth of *M. oryzae* was inhibited by WT JP2-270 and the Δ*bysR* pBBR2pCS-*ambR2* mutant. (I) Mycelial growth of *M. oryzae* was inhibited by WT JP2-270 and the Δ*bysR* pBBR2pCS mutant. (J) Histogram of mycelium growth radii. Data are means and standard deviations of quintuplicates. **, *P* < 0.01 (extremely significant difference compared with the WT).

**TABLE 2** Putative genes identified in the *ocf* gene cluster in *Burkholderia* sp. strain JP2-270

| Chr3 position[a] | Length (bp) | Predicted function[b] | Homolog[c] | Identity (%)[d] |
|---|---|---|---|---|
| 149511–151259 | 1,749 | FAD-linked oxidase domain protein | ORF1 (ACN32485.1) | 91.96 |
| 147277–148110 | 834 | LuxR-type regulator | *ambR1* (ACN32486.1) | 91.77 |
| 146137–146964 | 828 | LuxR-type regulator | *ambR2* (ACI01437.2) | 89.07 |
| 145346–145921 | 576 | Hypothetical protein | | |
| 143927–145627 | 1,701 | Cyclic peptide transporter | *ocfA* (ACJ24909.2) | 88.28 |
| 143431–143910 | 480 | Hypothetical protein | *ocfB* (ACL81525.1) | 89.38 |
| 133213–142701 | 9,502 | NRPS | *ocfD* (ACL81527.1) | 89.06 |
| 124072–133125 | 9,054 | NRPS | *ocfE* (ACL81528.1) | 91.89 |
| 120119–124033 | 3,915 | NRPS | *ocfF* (ACN32487.1) | 94.03 |
| 118515–120131 | 1,617 | Hydroxylase | *ocfG* (ACN32488.1) | 95.86 |
| 105109–109204 | 4,096 | Hybrid NRPS-PKS | *ocfH* (ACN32489.1) | 92.55 |
| 110901–114608 | 3,708 | Hybrid NRPS-PKS | *ocfH* (ACN32489.1) | 93.52 |
| 114605–118504 | 3,900 | Hybrid NRPS-PKS | *ocfH* (ACN32489.1) | 94.04 |
| 101789–105112 | 3,324 | Flavin-dependent monooxygenase | *ocfI* (ADT64845.1) | 92.42 |
| 97377–101472 | 4,096 | NRPS | *ocfJ* (ADT64846.1) | 91.51 |
| 96194–97153 | 972 | Halogenase | *ocfK* (ADT64847.1) | 91.02 |
| 94806–96116 | 1,311 | Transaminase | *ocfL* (ADT64848.1) | 92.37 |
| 93579–94607 | 1,029 | Epimerase | *ocfM* (ADT64849.1) | 94.39 |
| 92780–93499 | 720 | Thioesterase | *ocfN* (ADT64850.1) | 91.53 |

[a]Position in the genome of strain JP2-270 (GenBank no. CP029826).
[b]Predicted functions are based on annotation of *Burkholderia contaminans* MS14 (12).
[c]Homolog of the putative protein of *Burkholderia contaminans* MS14 (GenBank no. EU938698).
[d]Identity values were based on BLASTn results from the *Burkholderia* Genome DB.

A3-2-2 (Fig. 5A). Similarly, *Burkholderia stabilis* and *Burkholderia anthina*, clustered in subclade B3-2 (Fig. 5B), were located in subclades A3 and A3-2-1 (Fig. 5A), respectively. Comparative phylogenetic analysis provided evidence that the *ocf* gene cluster was transferred between some species, such as *B. contaminans* and *B. vietnamiensis*, *B. stabilis*, and *B. anthina*. Otherwise, the *ocf* gene cluster descended vertically within the genus. Thus, horizontal and vertical transfer both drive *ocf* gene cluster evolution.

In detail, we compared the *ocf* genes between *Burkholderia* sp. strain JP2-270 and *B. contaminans* MS14 and found that *ocfC*, encoding glycosyl transferase, was absent in the *ocf* gene cluster of JP2-270 (Fig. 5C), while a gene encoding a hypothetical protein was predicted between *ocfA* and *ambR2* in the JP2-270 *ocf* gene cluster (Fig. 5C and Table 2). In addition, the *ocfH* gene in MS14 was highly homologous with three independent genes, DM992_33365, DM992_33360, and DM992_33355, in JP2-270 (Fig. 5C and Table 2). The other genes of the JP2-270 *ocf* gene cluster are the same as those of *B. contaminans* MS14, with amino acid identity ranging from 88.28% to 95.86% (Table 2). The typical −35 and −10 boxes were predicted in the regions upstream of *ocfN*, *ocfJ*, *ambR2*, and *ambR1* in JP2-270, indicating that the putative promoters exist upstream of the respective genes (Fig. 6E). Notably, the putative promoters of *ocfN* and *ocfJ* were not identified in MS14, but a promoter was predicted upstream of *ocfL* in MS14 (10). Overall, the genes in the *ocf* gene cluster of JP2-270 have relatively high similarity with the corresponding genes in MS14 (Fig. 5C; Table 2).

**AmbR1 is a downstream component of the BysR regulation system and is directly regulated by BysR.** It is known that *ambR1* is a pathway-specific regulatory gene which positively regulates occidiofungin production. As the Δ*ocfE* mutant displays significantly decreased inhibitory activity against *M. oryzae* compared to JP2-270 and the expression of *ambR1* is significantly downregulated in the Δ*bysR* mutant (Fig. 1B, 2A, and 4F), we inferred that BysR might regulate occidiofungin production by modulating *ambR1* expression. Therefore, we enhanced the expression of *ambR1* in the Δ*bysR* mutant by introducing the *ambR1* expression vector controlled by a constitutive promoter, which resulted in increased suppressive activity against *M. oryzae* (Fig. 4G and J). The observed radius of mycelial growth was 0.86 ± 0.15 cm for the Δ*bysR* pBBR2pCS-*ambR1* strain and 0.72 ± 0.08 cm for JP2-270, indicating that *ambR1* could almost fully restore the inhibitory activity of the Δ*bysR* mutant (Fig. 4G and J). However, similar to pBBR2pCS, which was used as the control, overexpression of *ambR2* could not recover the inhibitory

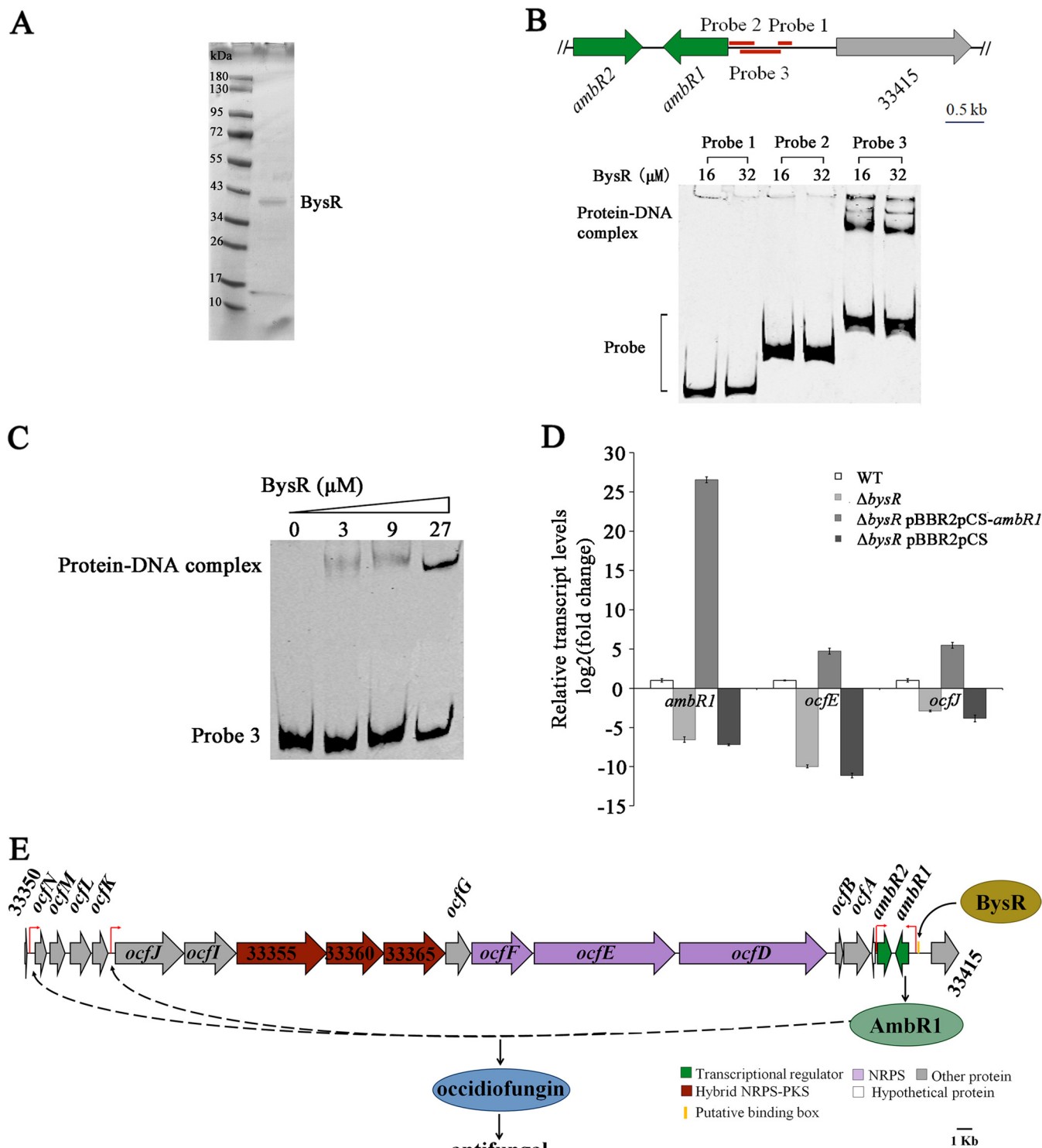

**FIG 6** Putative mode BysR regulation of occidiofungin synthesis. (A) SDS-PAGE of purified BysR protein. (B) Relative positions of different probe fragments in the *ambR1* promoter region and binding of the probes to BysR, determined by EMSA. (C) EMSA of BysR binding to probe 3 of the *ambR1* promoter region with increasing concentrations of BysR. (D) Expression levels of *ambR1*, *ocfE*, and *ocfJ* in WT JP2-270 and the Δ*bysR*, Δ*bysR* pBBR2pCS-*ambR1*, and Δ*bysR* pBBR2pCS mutants. The *recA* gene was used as a control. Gene expression levels are presented as expression ratios of the indicated genes. (E) Proposed model of how BysR controls the production of occidiofungin by directly activating the transcription of *ambR1*. The positions and orientations of the known genes are shown with horizontal arrows. The red arrows indicate putative promoter sequences. The black curved arrow indicates positive regulation. The dotted black curved arrows indicate predicted positive regulation.

activity of the Δ*bysR* strain (Fig. 4H to J). These results suggested that *ambR1* but not *ambR2* is a downstream component of the BysR system in JP2-270.

To further assess whether transcriptional regulation of *ambR1* was mediated by direct binding of BysR to the promoter, we performed electrophoretic mobility shift assays (EMSAs). BysR fused to an N-terminal glutathione *S*-transferase (GST) tag was purified using affinity chromatography, and then an ~39-kDa BysR protein was obtained by excising the GST tag (Fig. 6A). Three DNA fragments of the 5′-Cy5-labeled *ambR1* promoter, obtained by PCR amplification, were used as the probes. The probe with a fragment with positions −677 to −844 relative to the translation initiation site of *ambR1* was designated probe 1 (−677 to −844) (Fig. 6B). Similarity, the other two probes were named probe 2 (−7 to −336) and probe 3 (−160 to −692) (Fig. 6B). The EMSAs showed that the complex of BysR and probe 3 migrated more slowly than the unbound probe, while there were no shift bands observed for probe 1 and probe 2 (Fig. 6B). Moreover, probe 3 binding to BysR was significantly enhanced with increasing amounts of BysR protein (Fig. 6C). These results indicated that BysR could bind to the promoter region of *ambR1* and that the promoter region from −160 to −692 is essential for binding with BysR. Bioinformatics analysis also revealed that the consensus binding sequence with an AT-$N_{11}$-AT motif (5′-**AT**CGGCGATTTTC**AT**-3′) was present in probe 3 but not in probes 1 and 2 (Fig. S6). These results confirmed that BysR could bind to the promoter region of *ambR1* and that the consensus binding sequence is required for binding with BysR. In conclusion, BysR positively regulates the expression of *ambR1* by directly binding to its promoter.

**BysR regulates occidiofungin biosynthesis at the transcriptional level.** To clarify the regulatory effect of BysR on the *ocf* operon, we used qRT-PCR analysis to assess the transcriptional changes of *ocfJ*, *ocfE*, and *ambR1* in wild-type JP2-270 and the Δ*bysR*, Δ*bysR* pBBR2pCS-*ambR1*, and Δ*bysR* pBBR2pCS mutants. As shown in Fig. 6D, the deficiency of *bysR* resulted in a 7.2- to 28.6-fold reduction in the transcriptional levels of these three selected genes, relative to wild-type JP2-270. Moreover, the overexpression of *ambR1* in the Δ*bysR* mutant increased the expression levels of *ocfJ* and *ocfE*, resulting in 5.45-fold and 4.79-fold upregulation of *ocfJ* and *ocfE*, respectively, relative to the wild type (WT) (Fig. 6D), while the expression levels of *ocfJ*, *ocfE*, and *ambR1* in the Δ*bysR* pBBR2pCS strain were similar to those in wild-type JP2-270. The 26.56-fold-upregulated *ambR1* transcriptional level was observed in the Δ*bysR* strain harboring pBBR2pCS-*ambR1* (Fig. 6D), suggesting that *ambR1* was successfully overexpressed in the Δ*bysR* strain.

Based on our results, we concluded that (i) BysR binds directly to the promoter region of *ambR1* and activates the transcription of *ambR1* and (ii) AmbR1 promotes transcription of *ocf* genes located downstream of *ambR1* and thus controls the production of occidiofungin (Fig. 6E).

## DISCUSSION

**Occidiofungin has promising applications in agriculture and biomedicine.** Occidiofungin is a natural product produced by *Burkholderia* spp. and possesses broad-spectrum antifungal, antiparasitic, and anticancer activities with limited toxicity and chemical stability (2, 9, 30–32). Besides its marked potential for application in agricultural practices for controlling fungal plant diseases, occidiofungin has been developed as a lead compound for clinical antifungal therapeutics (e.g., candidiasis) and antiparasitic treatment owing to its distinctive mode of action (2, 9, 31, 32). In addition, occidiofungin and its derivatives could also be developed as anticancer drugs due to their targeting effect on some cancer cell lines (30). However, the lack of knowledge of the regulatory mechanisms of occidiofungin production significantly hinders effective utilization of the occidiofungin-producing strains, such as identification of highly productive strains and manipulation of the occidiofungin production pathway.

Our studies have shown that BysR was required in secondary metabolism and antagonism (14). Here, the antifungal activity of three metabolites produced by JP2-270 was analyzed, and we found that occidiofungin and pyrrolnitrin play important roles in inhibiting *M. oryzae* and *R. solani*, respectively (Fig. 2B and 4). Gladiostatin, recently

identified from *Burkholderia gladioli*, is a novel glutarimide antibiotic with promising anticancer activity (33). However, the gladiostatin-like compound produced by JP2-270 was not as effective as occidiofungin and pyrrolnitrin in inhibiting fungi (Fig. 2B). In order to study and make full use of microbial secondary metabolites for plant disease control, it is necessary to conduct research on the synthesis and regulation of secondary metabolites with potential as biological control agents.

The *ocf* gene cluster was found in 9 species of *Burkholderia* but not in any other genus, implying that vertical transmission is the main evolutionary mode of the *ocf* gene cluster within this genus. However, the *ocf* gene cluster was transferred in these 9 species (Fig. 5A and B), indicating that there were both horizontal and vertical transfers in the acquisition of this gene cluster in these species. The *ocf* gene cluster of JP2-270 was slightly different from that of *B. contaminans* MS14, such as *ocfC* (Fig. 5C). The absence of *ocfC* implies that the occidiofungin-like compound produced by JP2-270 may not have a glycosylation modification and is a new derivative of occidiofungin.

**BysR, as a global transcriptional regulator, regulates various biological processes.** LTTRs are multifunctional transcription regulators in prokaryotes and play an important role in carbon catabolism, amino acid metabolism, antibiotic resistance and motility, etc. (15, 18, 19, 21). BysR has been identified as a member of the LTTR family and highly homologous to Bcal3178, which controls biofilm formation and protease production (34).

In this study, the isolated beneficial strain *Burkholderia* sp. strain JP2-270 was used as a model. We identified the genes and associated pathways directly or indirectly regulated by BysR by exploring the RNA-seq and DAP-seq results. Previous studies suggested that *ambR1* and *ambR2* upregulated the expression of *ocfD* to *ocfJ*, which are responsible for occidiofungin biosynthesis (10, 11, 13, 35). Our results indicated that *ambR1* but not *ambR2* is a direct downstream target gene of BysR. Interestingly, BysR might directly regulate the expression of *ocfJ* and *ocfL*, as revealed by DAP-seq (Fig. 3E; Tables S4 and S5). These results suggested more complex and well-tuned regulation maps of occidiofungin production in *Burkholderia*. Of note, diaminobutyrate-2-oxoglutarate transaminase (DABAT; EC 2.6.1.76), encoded by the gene *ocfL* (DM992_33335), presumably catalyzes the formation of L-2,4-diaminobutyric acid (DABA) and 2-oxoglutarate (10, 36). DABA was added to the intermediates of occidiofungin production by OcfE (10), while 2-oxoglutarate may participate in the TCA cycle. The *ocfL* may not only provide aminobutyric acid during the synthesis of occidiofungin but also supplement the source of 2-oxoglutarate in the TCA cycle. Thus, we predicted that *ocfL* might be involved in both primary and secondary metabolic processes.

RNA-seq results showed that the genes associated with various cellular processes were regulated by BysR, among which the genes related to flagellar assembly and bacterial chemotaxis constitute the two most enriched pathways in the KEGG analysis (Fig. 1C). Moreover, the Lrp family transcriptional factor and H-NS histone family proteins are known as regulators modulating flagellar production (37), and their encoding genes were directly regulated by BysR in our DAP-seq (Fig. 3E; Table S4), implying that BysR might indirectly regulate flagellar assembly by targeting Lrp or H-NS genes. The previous study showed that *ocf* genes (including *ocfA* to *N* and *ambR2*), flagellar assembly-related genes (including *fliT*, *fliD*, and *motB*), and the genes associated with the type VI secretion system (including *hcp*, *tssC*, and *tssB*) were significantly downregulated in an *ambR1* mutant (MS455MT38) compared to the wild-type *Burkholderia* sp. strain MS455, as revealed by RNA-seq analysis (2). The results are consistent with our study showing that a *bysR*-defective mutant has altered expression of *ambR1*, which in turn affects the expression of the *ocf* gene cluster, flagellar assembly genes, and type VI secretion system-related genes. Therefore, integrating previous studies and our results, we inferred that BysR is involved in various cellular processes, including secondary metabolites production, TCA cycle, flagellar assembly, bacterial chemotaxis, and secretion.

Most remarkably, the genes encoding transcriptional regulators constitute the largest group directly regulated by BysR (Fig. 3E), such as LTTRs, LuxR family regulators, MarR family regulators, H-NS family regulators, IclR regulators, GntR family regulators,

Lrp/AsnC family regulators, TerR/AcrR family regulators, and so on. Obviously, BysR is the upstream regulator of other transcriptional factors. Therefore, in addition to the regulatory mechanisms related to secondary metabolite synthesis, we need more studies to identify other functions of BysR. Revealing more biological functions of BysR will provide a better understanding of how Bcc isolates adapt to environmental conditions.

**The feature of BysR regulation.** As an alternative and complementary method to chromatin immunoprecipitation sequencing (ChIP-seq), DAP-seq is powerful and convenient. As DAP-seq is not limited to one specific growth condition, it usually can reveal the binding events under conditions that are not suitable for ChIP-seq (26). DAP-seq has successfully been applied to identify transcription factor binding sites (TFBSs) in eukaryotes and prokaryotes (26, 38–41). In this study, DAP-seq was used to identify the TFBSs of BysR in the genome of JP2-270, and 400 putative binding loci were uncovered (Table S4). Based on the sequences of BysR binding loci, the consensus binding motif $AT-N_{11}-AT$ was predicted, which also displays the dyad symmetry pattern and has the typical $T-N_{11}-A$ motif of LTTRs (18). Although the classical LTTR binding locus is a dyad symmetry consensus $T-N_{11}-A$ motif (18), the binding sequences of LTTRs can vary significantly in base composition and length, such as the $T-N_{11}-A$ motif, which is found mainly in *Pseudomonas* (42), the $ATC-N_9-GAT$ motif in *Rhizobium* (43), and the $TTA-N_7-TAA$ motif in *Lactobacillus plantarum* (44). Furthermore, bioinformatics analysis revealed that the promoter region of *ambR1* possesses a few putative BysR binding loci (matching the consensus in Fig. 3D and Table S5), with a potential site (**AT**CGGCGATTTTC**AT**) downstream of the predicted −10 promoter region of *ambR1*, which is located close to the BysR DAP-seq peak summit (Merged-Chr3-148813-2) (Fig. S6). In addition, BysR binds to a variety of sites in JP2-270 (Table S4), and the majority of BysR binding loci (about 71%) contain the $AT-N_{11}-AT$ box (Table S5). This confirmed that the predicted consensus motif $AT-N_{11}-AT$ was the typical binding feature of BysR.

Based on our results, we summarized the biological processes that are potentially regulated by BysR and proposed a model of BysR-regulated occidiofungin synthesis (Fig. 3E and 6E). BysR not only regulates secondary metabolites but also participates in the regulation of various core metabolic pathways, such as the TCA cycle, amino acid synthesis and metabolism, and DNA replication. BysR is responsible for regulating the expression of multiple transcription factors, and it is possible that BysR occupies a higher regulatory position in the regulatory network of *Burkholderia* sp. strain JP2-270. For example, *ambR1*, as a downstream target gene, was confirmed to be directly regulated by BysR, and AmbR1 was reported to regulate flagellar assembly and the type VI secretion system in addition to *ocf* gene cluster expression.

## MATERIALS AND METHODS

**Bacterial/fungal strains and culture conditions.** Bacterial strains, fungal strains, and plasmids used in this study are listed in Table 3. *Burkholderia* sp. strain JP2-270 and derivates were routinely cultured in Luria-Bertani (LB) medium at 28°C (45). *Escherichia coli* strains were routinely maintained in LB medium at 37°C. The concentrations of antibiotics, when necessary, used for *E. coli* were 50 $\mu$g/mL for carbenicillin, 50 $\mu$g/mL for kanamycin, 50 $\mu$g/mL for streptomycin, and 30 $\mu$g/mL for nalidixic acid. *M. oryzae* isolate Guy11 and *R. solani* GD118 were routinely cultivated on complete medium (CM) (46) and potato dextrose agar (PDA) plates (200 g potato, 20 g glucose, 20 g agar, 1 L water; natural pH), respectively, at 25°C.

***In vitro* inhibition assay.** The petri dish assay was used to test *in vitro* antagonistic activity of JP2-270 and derivates against *M. oryzae* Guy11 and *R. solani* GD118. Overnight cultures of JP2-270 and derivates were streak inoculated at both sides, 2 cm away from the center of CM or PDA plates. After incubation for 24 h at 25°C, mycelium plugs (5-mm-diameter) from the fresh edge of *M. oryzae* Guy11 or *R. solani* GD118 were placed at the center of the CM (for Guy11) and PDA (for GD118) medium plates. Following 2 to 3 days for GD118 and 7 to 10 days for Guy11 coculture at 25°C, the radii of mycelium growth were measured to evaluate the inhibitory effect. Five biological replicates were performed, and an average value was calculated to evaluate the inhibitory activity.

**Growth curve.** The overnight cultures (15 h at 30°C) in LB were inoculated into 100 mL of rich medium (LB medium) (45) at a ratio of 1:100 and incubated at 30°C with shaking at 200 rpm. Sampling was performed at appropriate intervals up to 72 h according to the growth conditions. The growth rates were determined by measuring absorbance at 600 nm. The results reported here are averages for three replicate samples.

**TABLE 3** Bacterial strains and plasmids used in this study

| Strain or plasmid | Relevant characteristics | Source or reference |
|---|---|---|
| *Escherichia coli* | | |
| BL21(DE3) | F⁻ *ompT hsdS*$_B$(r$_B$⁻ m$_B$⁻) *gal dcm*(DE3) | Sangon Biotech, Shanghai, China |
| DH5α | F⁻ φ80*lacZ*ΔM15 Δ(*lacZYA-argF*)U169 *endA1 recA1 hsdR17*(r$_K$⁻ m$_K$⁺) *supE44 λ-thi-1 gyrA96 relA1 phoA* | Bethesda Research |
| | | |
| *Burkholderia* sp. | | |
| JP2-270 | Wild-type strain, NalR | This study |
| Δ*bysR* mutant | JP2-270 with 798-bp deletion in *bysR* | 14 |
| Δ*bysR* pBBR2-*bysR* mutant | Δ*bysR* carrying pBBR1MCS-2-*bysR* | 14 |
| Δ*bysR* pBBR2 mutant | Δ*bysR* carrying pBBR1MCS-2 | 14 |
| Δ*bysR* pBBR2pCS mutant | ΔbysR carrying pBBR1MCS-2pCS | This study |
| Δ*ocfE* mutant | JP2-270 with 3,264-bp deletion in *ocfE* | This study |
| Δ*prnC* mutant | JP2-270 with 1,419-bp deletion in *prnC* | This study |
| Δ*gdsE* mutant | JP2-270 with 1,665-bp deletion in *gdsE* | This study |
| Δ*bysR* pBBR2pCS-*ambR1* mutant | Δ*bysR* mutant carrying pBBR2pCS-*ambR1* | This study |
| Δ*bysR* pBBR2pCS-*ambR2* mutant | Δ*bysR* mutant carrying pBBR2pCS-*ambR2* | This study |
| | | |
| *Magnaporthe oryzae* Guy11 | Type strain | Fucheng Lin's lab |
| *Rhizoctonia solani* GD118 | Type strain | CNRRI |
| | | |
| Plasmids | | |
| pK18mobSacB | Insert detection plasmid; constructed by pK18, pSUP102 (RP4 mob) and *sacB*; Km$^r$; SacBS | ATCC |
| pk18mobSacBΔ*ocfE* | Carrying upstream and downstream homologous fragments of gene cluster *ocfE* | This study |
| pk18mobSacBΔ*prnC* | Carrying upstream and downstream homologous fragments of gene cluster *prnC* | This study |
| pk18mobSacBΔ*gdsE* | Carrying upstream and downstream homologous fragments of gene cluster *gdsE* | This study |
| pBBR1MCS-2 | Broad-host cloning vector, Km$^r$ | 29 |
| pBBR1MCS-2pCS | pBBR1MCS-2 carrying the promoter region of DM992_01545 | This study |
| pBBR2pCS-*ambR1* | pBBR1MCS-2pCS expressing the ambR1 gene of JP2-270 | This study |
| pBBR2pCS-*ambR2* | pBBR1MCS-2pCS expressing the ambR2 gene of JP2-270 | This study |

**Overexpression of *ambR1* and *ambR2* in the Δ*bysR* mutant.** The cold shock protein gene DM992_01545 (CP029824) was highly expressed in JP2-270 and the expression level was not affected by *bysR* mutation based on the RNA-seq results (GSE193778). The promoter region of DM992_01545 was amplified using pCS-F/pCS-R and cloned into the multiple-cloning site (MCS) of pBBR1MCS-2 to obtain the modified vector pBBR1MCS-2pCS. To construct a plasmid expressing *ambR1*, the fragment containing coding sequences of *ambR1* (DM992_33410) from the JP2-270 genome (CP029826) was cloned downstream of the constitutive promoter of pBBR1MCS-2pCS. The plasmid pBBR2pCS-*ambR1*, expressing *ambR1*, was obtained. The primers C*ambR1*-F and C*ambR1*-R were used to amplify the corresponding fragment. All constructs were verified by PCR and sequencing. Similarly, pBBR2pCS-*ambR2*, expressing *ambR2*, was constructed. The obtained plasmids were electroporated into the Δ*bysR* mutant to obtain the *ambR1*- and *ambR2*-overexpressing Δ*bysR* pBBR2pCS-*ambR1* and Δ*bysR* pBBR2pCS-*ambR2* strains, respectively. pBBR1MCS-2pCS was also transferred into the Δ*bysR* mutant as a negative control (Δ*bysR* pBBR2pCS strain). The primers used in this study are listed in Table S1.

**In-frame *ocfE*, *gdsE*, and *prnC* gene deletion in JP2-270.** Marker-free site-directed deletion was carried out using pK18mobSacB (47). The upstream and downstream regions of *ocfE*, *gdsE* and *prnC* to be deleted were fused using overlap extension PCR. The fusion products were then subcloned into the suicide vector pK18mobSacB. The resultant recombinant plasmids were introduced into JP2-270 by electroporation transformation, and subsequently, the plasmids were integrated into the target gene via homologous recombination. *Burkholderia* sp. strain JP2-270 is not sensitive to nalidixic acid. *Burkholderia* isolates containing the plasmids were selected on LB with 50 μg/mL kanamycin and 30 μg/mL nalidixic acid. The *Burkholderia* isolates containing plasmids were subsequently cultured in LB without any antibiotic for several generations. The deletion mutants that grew on LB plates with 30 μg/mL nalidixic acid but not on LB plates with 50 μg/mL kanamycin and 30 μg/mL nalidixic acid were selected, and the resultant *ocfE* (Δ*ocfE*), *gdsE* (Δ*gdsE*), and *prnC* (Δ*prnC*) allelic exchange mutants were verified by PCR and subsequent DNA sequencing (Fig. S7). The primers used in this study are listed in Table S1.

**RNA isolation, RNA-seq, and quantitative RT-PCR.** The single-colony bacteria of JP2-270 (WT) and the Δ*bysR* mutant (in-frame *bysR* deletion isolate) (14) were precultured in LB medium at 28°C overnight. The overnight cultures were inoculated into CM in a ratio of 1:100 and cultured at 28°C with shaking at 200 rpm. The cells were harvested after 24 h of incubation. Three biological replicates of each strain

were used. The collected cells were sent directly to Beijing Novogene Bioinformatics Technology Co., Ltd., for further treatments. Briefly, a total of 3 $\mu$g RNA per sample was obtained, and rRNA was depleted using a Ribo-Zero rRNA removal kit (Gram-negative bacteria). The mRNAs were fragmented using divalent cations at an elevated temperature in NEBNext first-strand synthesis reaction buffer (5×). Then, the cDNA libraries were generated using a NEBNext Ultra directional RNA library preparation kit for Illumina (New England Biolabs [NEB], USA) following the manufacturer's instructions. The cDNA fragments were purified with an AMPure XP system (Beckman Coulter, Beverly, MA, USA), and then 3 $\mu$L USER enzyme (NEB, USA) was used with size-selected, adaptor-ligated cDNA at 37°C for 15 min followed by 5 min at 95°C before PCR. Then, PCR was performed with Phusion high-fidelity DNA polymerase, universal PCR primers, and an index primer. Last, products were purified (AMPure XP system), and library quality was assessed on the Agilent Bioanalyzer 2100 system. The resultant samples were sequenced on an Illumina HiSeq 2000 platform.

For qRT-PCR analysis, an RNA preparation pure cell/bacterial kit (Tiangen Biotech, Beijing, China) was used to prepare total RNA according to the instructions. The quality and concentration were analyzed by 1% agarose gel electrophoresis. Genomic DNA (gDNA) was digested with gDNA remover (Toyobo) at 37°C for 5 min. Total RNA (~0.5 $\mu$g) was reverted to cDNA using a ReverTra Ace qRT-PCR master mix (Toyobo). Samples were diluted, about 25 ng of cDNA was added to the PCR system in a total volume of 20 $\mu$L, and qPCR was conducted using Thunderbird SYBR qPCR mix (Toyobo). The qRT-PCR was performed in an Applied Biosystems 7500 instrument according to the manufacturer's instructions. The *recA* gene was used as an internal standard, and relative expression was quantified using the $2^{-\Delta\Delta CT}$ threshold cycle ($C_T$) method. Primers are listed in Table S1. Three independent experiments were conducted, each with three replicates.

**DAP-seq analysis.** DAP-seq was carried out in duplicate on *Burkholderia* sp. strain JP2-270 genomic DNA as previously described (26). Specifically, the HaloTag-BysR protein was expressed using the TNT SP6 wheat germ protein expression system (Promega, Fitchburg, WI, USA). The HaloTag-BysR protein was purified using magnetic HaloTag beads (Promega) and verified by Western blotting with the anti-HaloTag antibody (Promega). Genomic DNA was extracted from JP2-270 using DNAiso (TaKaRa, Japan). A genomic DNA library was generated using a NEXTflex rapid DNA sequencing kit (Bioo Scientific, USA) following the manufacturer's instructions. The purified protein was incubated with DNA library at 30°C for 2 h, and then the unbound DNA fragments were washed away. Then, the BysR-bound DNA fragments were recovered and PCR amplified. The resultant products were sequenced using the Illumina NovaSeq platform with the PE150 sequencing strategy. Two technical duplicates were performed, and a DNA sample that did not undergo incubation with BysR was used as the input negative control.

**Bioinformatics analysis.** For RNA-seq data, clean reads were obtained by removing reads containing adapter, reads containing poly-N, and low-quality reads from raw data. Clean reads were mapped to the *Burkholderia* sp. strain JP2-270 reference genome (CP029824 to CP029828) using Bowtie 2-2.2.3 (48). Then, Htseq v0.6.1 was used to count the reads mapped to each gene (49). Measurement of FPKM (fragments per kilobase of transcript sequence per million base pairs sequenced) was used to estimate gene expression level (50). Differential expression analysis of two groups (three biological replicates per group) was performed using the DESeq R package (1.18.0). The square of the Pearson correlation coefficient ($R^2$) was calculated to estimate the reproducibility of the replicates. DESeq provides statistical routines for determining differential expression in digital gene expression data using a model based on the negative binomial distribution. The resulting $P$ values were adjusted using the Benjamini-Hochberg approach for controlling the false discovery rate. Genes with an adjusted $P$ value of <0.05 and |$\log_2$(fold change)| of >1 found by DESeq were assigned as differentially expressed. The volcano plot drawn by Origin 2022 was applied to display significantly different gene expression. KEGG (Kyoto Encyclopedia of Genes and Genomes) enrichment analysis was performed by KOBAS software (51).

For DAP-seq analysis, the clean reads were obtained by filtering the raw data using FASTP with the default parameters. The clean reads were mapped to the *Burkholderia* sp. strain JP2-270 reference genome (CP029824 to CP029828) using BWA-MEM (52). Peaks were generated with MACS2 (53) ($P$ value < 0.01), and IDR software was used to merge the peaks present in the two replicates. MEME-ChIP was used to analyze the conservative motifs in the peaks (54). The HOMER software (55) was used to annotate peaks. The box plots of the peaks' width distribution, the pie chart of distribution of BysR-bound genes, and the histogram distance to the TSS were drawn using Origin 2022. The Venn plot for RNA-seq and DAP-seq data was produced with Origin 2022.

Phylogenetic analysis was performed using phyloPhlAn 3 and MEGA7 software (56, 57). The open reading frames (ORFs) of the JP2-270 *ocf* gene cluster were predicted by the Softberry FGENESB program (Softberry, Inc., Mount Kisco, NY), and the identified ORFs were analyzed using a BLASTn search in the *Burkholderia* Genome Database (58). The Softberry BPROM program was used to identify the putative promoter sequences. The putative consensus binding sequences of BysR were identified by regular expression with the package in Python.

**Protein expression, purification, and EMSA.** The coding region of BysR was obtained by PCR with the primers BysR-F and BysR-R (Table S1) and cloned into vector pGEX-6P-1 to generate an N-terminally GST-tagged BysR protein fusion. The resulting plasmid was transformed into *E. coli* BL21(DE3) (Table 3) for protein expression. The resultant strain was cultivated in LB medium (containing 100 $\mu$g/mL ampicillin) overnight at 37°C. Two milliliters of the overnight culture was transferred into 200 mL fresh LB at 37°C and grown with shaking at 200 rpm, until an optical density at 600 nm ($OD_{600}$) of 0.4 to 0.6 was reached. Isopropyl-$\beta$-D-thiogalactopyranoside (IPTG; Sangon, Shanghai, China) was added to a final concentration of 0.4 mM. The culture was incubated for an additional 16 h at 18°C with shaking at 100 rpm. Cells were collected by centrifugation at 4°C, resuspended in 25 mL phosphate-buffered saline (PBS) lysis

buffer containing 10 mM protease inhibitor (phenylmethylsulfonyl fluoride [PMSF]; Beyotime, China), and lysed by sonication (JY92-IIDN; Scientz, Ningbo, China) (power, 200 W; ultrasonic, 5 s; intermittent, 10 s). Following centrifugation at 4,500 rpm at 4°C for 30 min, the soluble protein was collected by incubation with GST beads (Sangon, Shanghai, China) for 30 min. The purified GST-BysR protein was eluted with buffer containing glutathione. The GST tag was removed by PreScission protease according to the instructions (P2303; Beyotime, Shanghai, China). The purity of BysR protein was assessed by sodium dodecyl sulfate-polyacrylamide gel electrophoresis (SDS-PAGE), and the concentration of BysR was determined with a bicinchoninic acid (BCA) assay kit (Sangon Biotech, Shanghai, China). Aliquots of the proteins were stored at −80°C.

EMSAs were performed as follows. Cy5-labeled wild-type probes of the *ambR1* promoter region were obtained by PCR using EMSA-ambR1-F/EMSA-ambR1-R (probe 1; 330 bp), EMSA-ambR2-F/EMSA-ambR2-R (probe 2; 168 bp), and EMSA-ambR3-F/EMSA-ambR3-R (probe 3; 533 bp), respectively. The wild-type promoter DNA probes of genes DM992_38905 (probe 4; 272 bp) and DM992_31485 (probe 6; 268 bp) were amplified using EMSA-38905-F/EMSA-38905-R and EMSA-31485-F/EMSA-31485-R, respectively. The mutated nucleotides were introduced by primers (Table S1). The mutated probes for genes DM992_38905 (probe 5; 272 bp) and DM992_31485 (probe 7; 268 bp) were amplified by overlapping extension PCR using primers EMSA-38905-F/EMSA-MT38905-R and EMSA-MT38905-F/EMSA-38905-R and primers EMSA-31485-F/EMSA-MT31485-R and EMSA-MT31485-F/EMSA-31485-R, respectively. Probe and protein extract were mixed according to the specifications of the EMSA/gel shift binding buffer (5×) (GS005; Beyotime, Shanghai, China) and incubated for 20 min at room temperature in darkness. The binding mixture was loaded onto the 6% nondenatured polyacrylamide gel and electrophoresed at 100 V for 4 h in the dark. An Odyssey CLx infrared fluorescence imaging system (LI-COR, Lincoln, NE, USA) was used to detect the fluorescence signal of Cy5-labeled DNA fragments. Probes used are listed in Table S1.

**Statistics.** Data analysis was performed using Microsoft Excel 2010. Student's $t$ test was used to compare the differences between two sets of data. The differences between results were considered statistically significant when $P$ was $<0.05$ and extremely significant when $P$ was $<0.01$.

**Data availability.** Raw sequence data from our RNA-seq and DAP-seq analyses can be accessed via the National Center for Biotechnology Information Sequence Read Archive server under accession numbers GSE193778 and GSE193916, respectively, and the complete genome sequence of *Burkholderia* sp. strain JP2-270 has been deposited in NCBI under accession numbers CP029824 to CP029828.

## SUPPLEMENTAL MATERIAL

Supplemental material is available online only.

**SUPPLEMENTAL FILE 1**, PDF file, 2.2 MB.

## ACKNOWLEDGMENTS

This research was financially supported by National Natural Science Foundation of China (NSFC) (31901924 and 31972318) and supported by Zhejiang Provincial Key Research and Development Program of China (2021C02056-3), Zhejiang Provincial Natural Science Foundation of China (LY21C130004).

We thank Guozhong Feng (CNRRI, Hangzhou, China) for the platform support in the initial stage of this research. We thank Changfu Tian (College of Biological Sciences, China Agricultural University, Beijing, China) for his constructive comments on the manuscript.

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
