## [Reviewer comments · Microbiology Spectrum]

Microbiology Spectrum

BysR, a LysR-type Pleiotropic Regulator, Controls Production of Occidiofungin by Activating the LuxR-type transcriptional regulator AmbR1 in *Burkholderia* sp. JP2-270

Wu Lijuan, Liqun Tang, Yuchang He, Cong Han, Lei Wang, Yunzeng Zhang, and Zhiguo E

Corresponding Authors: Yunzeng Zhang, Yangzhou University, and Zhiguo E, China National Rice Research Institute

Review Timeline:

Submission Date:	July 13, 2022
Editorial Decision:	September 22, 2022
Revision Received:	November 22, 2022
Editorial Decision:	January 19, 2023
Revision Received:	February 18, 2023
Accepted:	February 24, 2023

Editor: Montarop Yamabhai

Reviewer(s): The reviewers have opted to remain anonymous.

Transaction Report:

DOI: <https://doi.org/10.1128/spectrum.02684-22>

September 22, 2022

Dr. Zhiguo E
China National Rice Research Institute
Hangzhou
China

Re: Spectrum02684-22 (BysR, a LysR-type Pleiotropic Regulator, Controls Production of Occidiofungin by Activating the LuxR-type transcriptional regulator AmbR1 in Burkholderia sp. JP2-270)

Dear Dr. Zhiguo E:

Thank you for submitting your manuscript to Microbiology Spectrum. I have received comments from 2 experts in the field. Both of the reviewers and I agree that this manuscript requires major revisions before it can be published.

The comments of the reviewers to the editor are as follows.

Many of the authors' statements are not supported by the bioinformatics data presented. The bioinformatics were not performed well and were often misleading (Figures 1, 4, and 6). Much of the bioinformatics should either be removed from the paper or redone. In addition, extensive corrections need to be made for English grammar. There are probably grammatical errors in more than half the sentences in the paper. Some experiments of EMSA concerning AmbR1 and its targets would be useful. The model presented in Fig. 7 is wrong the arrows referred to AmbR1 indicate that it is an activator of transcription acting on a site, not mentioned in the paper, that seems to be a transcription start site or a promoter directed toward the direction of the arrows indicating the ocf ORFs. Presumably AmbR1/2 act on site/s located upstream the ocf cluster ORFs. Therefore, major revision is necessary.

While we are willing to consider a revised version of this paper at Spectrum, it is highly important to improve the writing. I recommend that you ask a colleague of yours who is a native English speaker to read and provide you some feedback on the writing. You are also welcome to use one of the services here: <https://journals.asm.org/content/language-editing-services>

Link Not Available

Sincerely,

Montarop Yamabhai

Journals Department
Reviewer comments:

Reviewer #1 (Comments for the Author):

In this paper, the authors studied a Burkholderia LysR-type regulator called BysR that they had previously found to be important for production of the antifungal compound occidiofungin. RNAseq was performed to identify genes affected by a *bysR* deletion, and DAPseq was used to identify binding sites for BysR. They found that RNA levels of occidiofungin and other secondary metabolites were affected, as well as a variety of other pathways. Overexpression of *ambR1* on a plasmid bypassed the need for BysR, which suggested that BysR acts upstream of *AmbR1*. Purified BysR was able to directly bind to *ambR1* promoter DNA, which implies that BysR regulates occidiofungin synthesis by binding and activating the *ambR1* promoter.

The RNAseq, DAPseq, plasmid expression, and EMSA experiments may contribute to our understanding of the regulation of secondary metabolites in Burkholderia species. However, much of the paper was devoted to bioinformatic analysis that was poorly performed. As described in the points below, phylogenetic trees seemed to be generated from random unrelated sequences or by using a poor choice of algorithms. In addition, a binding motif identified for BysR is probably a spurious motif that has nothing to do with BysR. Finally, there are likely grammatical errors in more than half the sentences of this paper. There are also spelling errors including: harbours, ploy-N, significantly, plasys, envolved, and encoding.

1. It is unclear how the protein sequences in Figure 1A were chosen. Most sequences appear to be unrelated. For example, the *E. coli* CynR is closest to the *Staphylococcus aureus* CidR in the tree, and the closest non-Burkholderia homolog to BysR is from *Vibrio cholerae*. The long branches indicate a lack of homology. A quick BLAST search shows many close Burkholderia BysR homologs, which contradicts the assertion on line 98. The authors need to create a new tree with a better selection of BysR homologs.
2. The authors claim "great dispersion" between the wildtype and Δ *bysR* RNAseq samples on line 113, but Figures 2A and S1 show a lot of similarity.
3. The authors claim "high consistence between the RNAseq and qRT-PCR results" on line 116, but Figure S2 shows only a minimum of consistency between the RNA-seq and qRT-PCR results.
4. The authors state "3.6-7.9 folds" downregulated on line 123, but it should be about 100-fold or more downregulated from Figure 2B. The authors need to convert from the log scale when referring to fold changes. There are similar problems on line 217 and throughout the paper.
5. On line 119 and in Figure 2B, a p-value threshold of 0.05 is indicated, but a p-value threshold of 0.01 is given on line 703 of the figure legend.
6. In Figure 2D, FlgE should point to the hook. FliG should point to the yellow blob above FliM. Rather than "Cytoplasmic membrane", FlgN, FliJ, FliS, and FliT are secretion chaperones. FlgA helps assembly of the P ring in the periplasm. The rod cap, FlgJ, is missing from the figure. For example, see Evans, Hughes, and Fraser. 2014. Trends in Microbiology 22(10):566-572.
7. In Figure 4B and C, the authors need to explain the acronyms TSS, TTS, and NR.
8. On line 164, 93% of the DAP-seq peaks are in the -1 kbp to 1 kbp regions. The authors need to put this in context: how much of the rest of the genome is in this region? (I would imagine most of it.)
9. The LTTR box motif of "TN11A" on line 167 is remarkably low information and ignores most of the information content in Figure 4D. LTTR boxes are probably palindromic, but this motif is not. There seems to be a pattern of SW-SW-SW-SW, and the three base repeat suggests a pattern within the coding sequence. Did this pattern actually match to sequences in promoter regions? If not, this pattern might not have anything to do with BysR or LTTR boxes. The authors need to provide additional evidence supporting this motif.
10. In Figures 5 and 7D, the genotypes should be correctly stated: Δ *bysR* pBBR2-*ambR1* (not *ambR1+*).
11. In Figure 6A and B, the authors do not rule out that a common ancestor had the occidiofungin operon and that different species lost the operon. The authors should redo Figure 6 using parsimony (instead of neighbor-joining) to better show the phylogenetic relationships, which should help to distinguish these possibilities.
12. There are no references or compositions for CM medium or PDA plates on lines 404-405.

Reviewer #2 (Comments for the Author):

Even if the paper adds important clues about the regulation of Occidiofungin production in the context of the role of BysR regulon functions, involved in other secondary metabolites production, motility and chemotaxis of Burkholderia, it needs a deep revision.

The main concern is the quality and clarity of the text that should be improved to make the paper clear and easily readable. The second important work to do, is about the precise annotation of the Burkholderia sp. JP2/270 genome in the regions object of the research: the ocf clusters to define genes and operon(s), and the sequences of the loci involved in BysR regulon as recognised by transcriptomic and DAP-seq results.

This analysis should concern:

1. Putative promoters localisation in Burkholderia sp. JP2/270 genome such as been done by Gu et al. AEM 2011 on the B.contaminas MS14 genome and comparison (the experimental transcription start determination of ocf and ambR1 and ambR2, should be useful but object of another paper..).
2. Authors in Fig.4D predict the putative binding site of BysR, but they don't check, for the presence of this consensus, the sequence of probe 3 (upstream ambR1) used for EMSA Fig. 7B and C, neither the sequences recognised by DAP-seq.
3. AmbR1/AmbR2 binding sites, and the molecular mechanism of transcription regulation of these LuxR like regulators are unknown, even if AmbR1 seems to be involved in the expression of many other genes besides ocf genes in another Burkholderia sp.(see Ref. 2 in the paper). Authors should investigate the AmbR1 binding sequences present in the surroundings of codon start of ocfL, ambR2 (see Ref 12, Gu et al. AEM, 2011) and perhaps ocfN putative promoters by EMSA tests.
4. Because of the absence of the analyses to be done in point 1 and 3, the model proposed in Fig7D is wrong and seems to suggest the presence of a mistake: an arrow suggests the presence of a transcription start site directed toward the translation direction of the ocf genes ORFs. In all organisms in traslatino the ORF reading goes from the start to the stop codons found from the 5' to the 3' end of the mRNA and 5' to 3' is also the direction of the transcription. Transcription and translation are codirected, and in bacteria coupled, in particular this aspect is important for transcription/translation regulation of operons.
5. Independently of the language quality the Discussion is confused and careless and the reasoning badly organised.

Staff Comments:

Preparing Revision Guidelines

Please return the manuscript within 60 days; if you cannot complete the modification within this time period, please contact me. If you do not wish to modify the manuscript and prefer to submit it to another journal, please notify me of your decision immediately so that the manuscript may be formally withdrawn from consideration by Microbiology Spectrum.

General Comments and Criticisms

This paper by means of genetic, transcriptomic, genomic and molecular biology approaches adds some important informations on the role of a recently recognised new Lys/R type LTTR transcriptional regulator, named BysR.

The authors investigate the positive role of BysR on the expression of the genomic island dedicated, in *Burkholderia* sp. JP2/270, to the synthesis of the well known antifungal oligopeptide Occidiofungin, active against the rice pathogen *Magnaporthe oryzae* or *grisea* causing the rice blast disease.

They identify by transcriptomic (RNA seq and qRT-PCR) the genes and operons target of the BysR positive or negative regulation that can be directly exerted, as seems the case of the *ambR1* gene for the specific regulator of the *ocf* gene cluster, or indirectly, as suggested by the comparison of transcriptomic and DAP-seq results.

The role of BysR on the expression of *ambR1* is well ascertained by genetics and the results of EMSA.

Even if the paper adds important clues about the regulation of Occidiofungin production in the context of the role of BysR regulon functions, involved in other secondary metabolites production, motility and chemotaxis of *Burkholderia*, it needs a deep revision.

The main concern is the quality and clarity of the text that should be improved to make the paper clear and easily readable.

The second important work to do, is about the precise annotation of the *Burkholderia* sp. JP2/270 genome in the regions object of the research: the *ocf* clusters to define genes and operon(s), and the sequences of the loci involved in BysR regulon as recognised by transcriptomic and DAP-seq results.

This analysis should concern:

1. Putative promoters localisation in *Burkholderia* sp. JP2/270 genome such as been done by Gu et al. AEM 2011 on the *B. contaminans* MS14 genome and comparison (the experimental transcription start determination of *ocf* and *ambR1* and *ambR2*, should be useful but object of another paper..).
2. Authors in Fig.4D predict the putative binding site of BysR, but they don't check, for the presence of this consensus, the sequence of probe 3 (upstream *ambR1*) used for EMSA Fig. 7B and C, neither the sequences recognised by DAP-seq.
3. AmbR1/AmbR2 binding sites, and the molecular mechanism of transcription regulation of these LuxR like regulators are unknown, even if AmbR1 seems to be involved in the expression of many other genes besides *ocf* genes in another *Burkholderia* sp.(see Ref. 2 in the paper). Authors should investigate the AmbR1 binding sequences present in the surroundings of codon start of *ocfL*, *ambR2* (see Ref 12, Gu et al. AEM, 2011) and perhaps *ocfN* putative promoters by EMSA tests.
4. Because of of the absence of the analyses to be done in point 1 and 3, the model proposed in Fig7D is wrong and seems to suggest the presence of a mistake: an arrow suggests the presence of a transcription start site directed toward the translation direction of the *ocf* genes ORFs. In all organisms in translation the ORF reading goes from the start to the stop codons found from the 5' to the 3' end of the mRNA and 5' to 3' is also the direction of the transcription. Transcription and translation are co-directed, and in bacteria coupled, in particular this aspect is important for transcription/translation regulation of operons.
5. Independently of the language quality the Discussion is confused and careless and the reasoning badly organised. This section needs to be re-written.

Minor Observations

- L. 42 Add: or *grisea*, to *Magnaporthe oryzae*, since actually there is confusion between the two names
- L. 87 Add: TCSs, means two components regulatory system
- Fig. 1 Legend, better: Structural domains of BysR and comparison to related LTTR proteins. In the 1B scheme "cofactor" is more indicated than "substrate"
- Fig. S1 Pearson
- Fig. S2 (and consequently L.115-116 and L.452-462) the RNA seq bars (the use of grey nuances renders difficult the recognition) are represented in the same scale of units than the qRT-PCR

results? Specify how were calculated the values of these bars and that for the qRT-PCR were analysed samples from the two strains: *wt* and Δ *bysR*.

- L. 120: delete with.
- L. 144-145 unclear sentence.
- L 146 Besides, BysR, not *bysR*, the protein function not the gene. Be careful, in other parts of the paper there are similar gene notation/function confusion, i.e. in Fig.7 in the green oval AmbR1 not *ambR1*.
- Table S5: Colours meaning?
- L. 170:locus not loci, but better is consensus sequences.
- L.416: pBBR1-MCS-2 reference or origin.
- L.212 add: is indicative.
- Fig.5 labels on the pictures and graph are confusing, better to use the corresponding precise descriptions of L. 745-746.
- Fig. 6 legend: L.749 remove based. L.752 Ocf protein or proteins? Specify the protein/proteins used to infer the phylogenetic tree based on proteins comparison from different sp. Explain the meaning of the coloured dotted lines.
- L.251-252: is not unusual that in bacterial NPRS , PKS or NPRS-PKS, different enzyme domains linearly coded from a large ORF such as MS14 *ocfH* are split in different ORFs in another strain or species, Authors should comment.
- L.254-263 the sentences should be re-written to become comprehensible
- L. 311-312 confusing sentence: in fact, Authors demonstrate that is BysR that regulates *ambR1* transcription, so is the regulation of AmbR1 that is downstream BysR..

Dear reviewers:

Thank you very much for giving us constructive suggestions on our manuscript entitled “BysR, a LysR-type Pleiotropic Regulator, Controls Production of Occidiofungin by Activating the LuxR-type transcriptional regulator AmbR1 in *Burkholderia* sp. JP2-270” (Spectrum02684-22). These insightful comments and suggestions would help us both in English and in depth to improve the quality of the paper.

We have studied reviewers’ comments carefully. According to the reviewers’ detailed suggestions, we have made a careful revision on the original manuscript. Below is our responses to the reviewers’ comments. The reviewer’s comments are in normal and our responses are in bold with line numbers referring to the revised manuscript.

Reviewer comments:

In this paper, the authors studied a *Burkholderia* LysR-type regulator called BysR that they had previously found to be important for production of the antifungal compound occidiofungin. RNAseq was performed to identify genes affected by a *bysR* deletion, and DAPseq was used to identify binding sites for BysR. They found that RNA levels of occidiofungin and other secondary metabolites were affected, as well as a variety of other pathways. Overexpression of *ambR1* on a plasmid bypassed the need for BysR, which suggested that BysR acts upstream of *AmbR1*. Purified BysR was able to directly bind to *ambR1* promoter DNA, which implies that BysR regulates occidiofungin synthesis by binding and activating the *ambR1* promoter.

The RNAseq, DAPseq, plasmid expression, and EMSA experiments may contribute to our understanding of the regulation of secondary metabolites in *Burkholderia* species. However, much of the paper was devoted to bioinformatic analysis that was poorly performed. As described in the points below, phylogenetic trees seemed to be generated from random unrelated sequences or by using a poor choice of algorithms. In addition, a binding motif identified for BysR is probably a spurious motif that has nothing to do with BysR. Finally, there are likely grammatical errors in more than half the sentences of this paper. There are also spelling errors including: harbore, ploy-N, significatly, plasys, envolved, and enconding.

1. It is unclear how the protein sequences in Figure 1A were chosen. Most sequences appear to be unrelated. For example, the *E. coli* CynR is closest to the *Staphylococcus aureus* CidR in the tree, and the closest non-*Burkholderia* homolog to BysR is from *Vibrio cholerae*. The long branches indicate a lack of homology. A quick BLAST search shows many close *Burkholderia* BysR homologs, which contradicts the assertion on line 98. The authors need to create a new tree with a better selection of BysR homologs.

We acknowledge this valuable comment. In the previous manuscript, the reported LTTR proteins from different bacteria species were included for phylogenetic tree construction, and the low sequence similarity of certain sequences probably caused the phylogenetic tree not reliable enough. Based on the reviewer’s suggestion, we re-selected the LTTR proteins of *Burkholderia* genus, and constructed the protein phylogenetic tree with OxyR from *E. coli* species (accession no. AAN83350.1) as the outgroup. The results were also redescrbed in Line 96-100 “The phylogenetic analysis of BysR with its most relevant LTTRs revealed that BysR is a novel LTTR within the genus *Burkholderia*, and the most homologous to BysR is BcaI3178 of *Burkholderia cenocepacia* H111, showing 93% identity (96% similarity) (Fig. 1A). The LTTRs associated with secondary metabolite synthesis are ScmR of *Burkholderia thailandensis* and ShvR of *Burkholderia cenocepacia* (20, 23) (Fig. 1A).” Please see Line 96-100 in the revised manuscript and Fig. 1A.

2. The authors claim "great dispersion" between the wildtype and Δ bysR RNAseq samples on line 113, but Figures 2A and S1 show a lot of similarity.

We agree the reviewer's comment. We sought to show that the samples from wildtype and Δ bysR strains formed two separate groups, and thus the RNA-seq results were reliable. However, the word "great dispersion" is not appropriate. In the revised manuscript, we have rewritten the sentence in Line 110-112 "Principal component analysis (PCA) revealed that three replicates of the WT group and the Δ bysR group are samples of high similarity and the samples of different groups could be clearly distinguished on the plot of PC1 and PC2." Please see Line 110-112 in the revised manuscript.

3. The authors claim "high consistence between the RNAseq and qRT-PCR results" on line 116, but Figure S2 shows only a minimum of consistency between the RNA-seq and qRT-PCR results.

According to the reviewer's valuable comment, we have re-described the results of Figure S2. Please see Line 114-119 "We further compared the relative expression level of random selected genes by qRT-PCR and RNA-seq, and it indicated that there are some variation in the magnitude of fold change, but the changing trends are the same (Supplementary Fig. S2). The consistence change trend and moderate correlation of gene expression levels between RNA-seq and qRT-PCR results further demonstrated the reliability of the RNA-seq results (Supplementary Fig. S2)." in the revised manuscript. The Figure S2 shows that the relative expression levels of these selected genes revealed by qRT-PCR exhibit consistent upregulated or downregulated trend as revealed by RNA-seq. The fold change inconsistencies revealed by qRT-PCR and RNA-seq are mainly due to the methods used for the two assays, and the inconsistencies are also frequently observed by previous studies, e.g. Coenye et al., 2021 *Biofilm*, Cao et al., 2021 *AEM*, Zhou et al., 2022 *Frontiers in Cellular and Infection Microbiology*.

4. The authors state "3.6-7.9 folds" downregulated on line 123, but it should be about 100-fold or more downregulated from Figure 2B. The authors need to convert from the log scale when referring to fold changes. There are similar problems on line 217 and throughout the paper.

Thanks for this valuable comment. We have convert the log scale throughtout the paper. The gene expression level described by FPKM value or by \log_2 (fold change) were described in detail in the corresponding positions in this paper and the "3.6-7.9 folds" had been re-write as 3.6-7.9 fold change in the revised manuscript, please see Line 124-127 "As shown in Fig. 3A, the FPKM values of genes related to occidiofungin synthesis in Δ bysR mutants were significantly lower than those in wild type JP2-270, and these genes were dramatically downregulated by 3.6-7.9 fold change in the Δ bysR mutants (Fig. 2B)." and Line 232-234 "As mentioned above, we found that the expression levels (base on FPKM value) of several genes involved in occidiofungin (Region 3.2), gladiostatatin (Region 3.7), and pyrrolnitrin (Region 3.6) biosynthesis were dramatically decreased in Δ bysR mutant (Fig. 3A, Supplementary Table S3)." in the revised manuscript.

In Fig. 2B, the x-coordinate represents the \log_2 (fold change) and y-coordinate represents the $-\log_{10}$ (p-value). The 3.6-7.9 folds downregulated on line 123 refers to the value of \log_2 (fold change).

5. On line 119 and in Figure 2B, a p-value threshold of 0.05 is indicated, but a p-value threshold of 0.01 is given on line 703 of the figure legend.

Sorry for this typo error. We have changed the p-value from 0.01 to 0.05. Please see Line 833 in the

revised manuscript.

6. In Figure 2D, FlgE should point to the hook. FliG should point to the yellow blob above FliM. Rather than "Cytoplasmic membrane", FlgN, FliJ, FliS, and FliT are secretion chaperones. FlgA helps assembly of the P ring in the periplasm. The rod cap, FlgJ, is missing from the figure. For example, see Evans, Hughes, and Fraser. 2014. Trends in Microbiology 22(10):566-572.

Thanks for the thoughtful advice. According to the reviewer's advice, the flagellar model was redrawn. Please see Figure 2D in the revised manuscript.

7. In Figure 4B and C, the authors need to explain the acronyms TSS, TTS, and NR.

According to the reviewer's valuable comment, we have annotated the acronyms of TSS, TTS and NR in the caption of Figure 4B and C. Please see Line 855 and 857 in the revised manuscript.

8. On line 164, 93% of the DAP-seq peaks are in the -1 kbp to 1 kbp regions. The authors need to put this in context: how much of the rest of the genome is in this region? (I would imagine most of it.)

We agree the reviewer's valuable comment. We have changed the regions from -1 kbp ~ 1 kbp to -700 bp ~ 100 bp and estimated the proportion of the DAP-seq peaks. Please see Line 167-169 "Among the identified peaks, 232 (58%) of these peaks were located in the -700 bp to 100 bp regions by the analysis of peak summit positions relative to the start codons of JP2-270 open reading frames" in the revised manuscript and Figure 4C.

9. The LTTR box motif of "TN11A" on line 167 is remarkably low information and ignores most of the information content in Figure 4D. LTTR boxes are probably palindromic, but this motif is not. There seems to be a pattern of SW-SW-SW-SW, and the three base repeat suggests a pattern within the coding sequence. Did this pattern actually match to sequences in promoter regions? If not, this pattern might not have anything to do with BysR or LTTR boxes. The authors need to provide additional evidence supporting this motif.

Many thanks to the reviewer's valuable comment. As mentioned by the reviewer, the motif of Fig. 4D in the original manuscript is quite different from that of classic LTTR box motif. After careful re-analysis of the data, we found that the motif predicted by MEME-Chip was based on all the 400 peaks obtained by sequencing. In the revised manuscript, all peaks are sorted out and only peaks with a length greater than 500 bp and located in the region -700 bp to 100 bp relative to the transcription start site are retained for MEME-Chip analysis. The resulting consensus binding sequence is AT-N₁₁-AT, which is similar to the motif of LTTR OxyR from *Escherichia coli*. Two peaks containing this consensus binding sequence were random selected for EMSA verification, and the results also showed that AT-N₁₁-AT box is required for binding of BysR. Detailed results are shown in line 170-187 "A total of 134 peaks, with a length greater than 500 bp and located in the region -700 bp to 100 bp relative to the transcription start site of the corresponding genes, were used for motif prediction using MEME-ChIP (26) and the conserved motif-like AT-N₁₁-AT box (e-value = 3.8e-021) was found in 95 out of 134 peaks (71%) (Fig. 4D and Supplementary Table S6). This conserved A+T rich box was similar to the LTTR OxyR consensus binding sequence (27) and the T-N₁₁-A motif, typical of LTTRs, was also recognized in this conserved box (28). Thus we proposed that the AT-N₁₁-AT box was the binding feature of BysR (Fig. 4D). Further, the potential BysR binding sequences in 95 peaks were

searched by regular expression module in python, and at least one AT-N₁₁-AT box sequence was found in each of these 95 peaks (Supplementary Table S6), implying that BysR binds essentially around the proposed consensus loci to regulate the transcription of the target genes. In order to confirm that the AT-N₁₁-AT box sequences are essential for BysR binding, the promoter region fragments of genes DM992_38905 and DM992_31485, with wild-type and mutated sequences of motif-like consensus sequences, were labeled with Cy5 and used for EMSA analysis. The presence of shift bands in the wild-type probes (probe 4 and probe 6) and the absence of BysR binding to the mutated probes (probe 5 and probe 7) indicate that the predicted consensus-binding sequences are necessary for BysR to bind to the promoter region and regulate the transcription of the targeted genes (Supplementary Fig. S3).” and Fig 4D, Supplementary Table S6, Fig S3 in the revised manuscript.

10. In Figures 5 and 7D, the genotypes should be correctly stated: Δ bysR pBBR2-ambR1 (not ambR1+).

According to the good advice, we have corrected the genotypes *ambR1+* to Δ *bysR* pBBR2-*ambR1*, *ambR2+* to Δ *bysR* pBBR2-*ambR2* and *bysR+* to Δ *bysR* pBBR2-*bysR*, please see Figure 5 and Figure 7D in the revised manuscript.

11. In Figure 6A and B, the authors do not rule out that a common ancestor had the occidiofungin operon and that different species lost the operon. The authors should redo Figure 6 using parsimony (instead of neighbor-joining) to better show the phylogenetic relationships, which should help to distinguish these possibilities.

According to the reviewer’s good advice, we have redo Figure 6 using maximum parsimony. Please see Figure 6 in the revised manuscript. The phylogenetic relationship between genome and Ocf protein was redescribed, and the detailed method of phylogenetic tree construction was also redescribed. Please see Line 260-271 “The phylogenetic tree was constructed based on the concatenated sequences of AmbR1/2 and OcfA-OcfN proteins (Fig. 6B). To investigate whether the evolution of Ocf protein coincides with strain evolution, we compared the phylogenetic relationships of Ocf proteins and *Burkholderia* genomes. The evolutionary trajectory of Ocf proteins from some species was inconsistent with the genome-wide evolution of the corresponding host strains. *B. contaminans* and *B. vietnamiensis* were clustered together in the subclade B3-1 (Fig. 6B), but they were distributed separately in A2 and A3-2-2 subclades (Fig. 6A). Similarly, *B. stabilis* and *B. anthina*, clustered in the B3-2 subclade, (Fig. 6B) were located in A3 and A3-2-1 subclades (Fig. 6A), respectively. Besides, *B. ubonensis* was the outermost part of the Ocf protein phylogenetic tree, while it was located in the A3-2 subclade in the genome-wide phylogenetic tree (Fig. 6A and B). All these results indicated that occidiofungin genes might be acquired via horizontal and vertical transfer.” and Line 878-885 “Phylogenetic tree inferred by MEGA7 using maximum parsimony (MP) method with tree-bisection-regrafting (TBR) algorithm (68). The most parsimonious tree with length= 45891 is shown. PhyloPhlAn 3 was used to select conserved marker genes encoding a total of 2,0092 amino acids in the genome sequences of representative species belonging to *Burkholderia*. The combined sequences of all marker genes were used for phylogenetic analysis. The numbers above branches are bootstrap support values > 60 % from 1000 replications, with an average branch support of 90%. (B) The phylogenetic tree of Ocf protein generated by MEGA7 (68). ” in the revised manuscript, respectively.

12. There are no references or compositions for CM medium or PDA plates on lines 404-405.

Thanks for the good advice, we have add the reference for CM medium and the compositions for PDA plates on Line 493-494“ *M. oryzae* isolate Guy11 and *R. solani* GD118 were routinely cultivated on complete medium (CM medium)(58) and PDA plates (potato 200 g, glucose 20 g, agar 20 g, water 1 L, natural pH), respectively, at 25 °C.” , please see the revised manuscript.

Reviewer #2 (Comments for the Author):

Even if the paper adds important clues about the regulation of Occidiofungin production in the context of the role of BysR regulon functions, involved in other secondary metabolites production, motility and chemotaxis of Burkholderia, it needs a deep revision.

The main concern is the quality and clarity of the text that should be improved to make the paper clear and easily readable.

The second important work to do, is about the precise annotation of the Burkholderia sp. JP2/270 genome in the regions object of the research: the *ocf* clusters to define genes and operon(s), and the sequences of the loci involved in BysR regulon as recognised by transcriptomic and DAP-seq results.

This analysis should concern:

1. Putative promoters localisation in Burkholderia sp. JP2/270 genome such as been done by Gu et al. AEM 2011 on the *B. contaminans* MS14 genome and comparison (the experimental transcription start determination of *ocf* and *ambR1* and *ambR2*, should be useful but object of another paper..).

Thanks for the thoughtful advice. The precise annotation of the *Burkholderia* sp. JP2-270 *ocf* gene clusters and putative promoters localisation were done according to the methods described by Gu et al. AEM 2011. Please see Line 272-283 “In detail, we compared the *ocf* genes between *Burkholderia* sp. JP2-270 and *B. contaminans* MS14, and found that *ocfC* encoding glycosyl transferase was absent in the *ocf* gene cluster of JP2-270 (Fig. 6C), while a gene encoding hypothetical protein was predicted between *ocfA* and *ambR2* in JP2-270 *ocf* gene cluster (Fig. 6C and Table 2). Besides, the *ocfH* in MS14 was highly homologous with three independent genes DM992_33365, DM992_33360, and DM992_33355 in JP2-270 (Fig. 6C and Table 2). The other genes of the JP2-270 *ocf* gene cluster are the same as those of the *B. contaminans* MS14, with the amino acid identity ranging from 88.28% to 95.86% (Table 2). The typical -35 and -10 boxes were predicted in the upstream of *ocfN*, *ocfJ*, *ambR2*, and *ambR1* in JP2-270, indicating that the putative promoters exist upstream of the respective genes; their locations are shown in Fig. 7E. Notably, the putative promoters of *ocfN* and *ocfJ* were not identified in MS14 but a promoter was predicted upstream of *ocfL* in MS14 (12). Overall, the genes in the *ocf* gene cluster of JP2-270 have relatively high similarity with the corresponding genes in MS14 (Fig. 6C, Table 2).” and Line 590-595 “The open reading frames (ORFs) of JP2-270 *ocf* gene cluster were predicted by Softberry FGENESB program (Softberry, Inc., Mount Kisco, NY), and the identified ORFs were analyzed using BLASTN search in Burkholderia Genome Database (35). The Softberry BPROM program was used to identify the putative promoter sequences.” in the revised

manuscript.

2. Authors in Fig.4D predict the putative binding site of BysR, but they don't check, for the presence of this consensus, the sequence of probe 3 (upstream *ambR1*) used for EMSA Fig. 7B and C, neither the sequences recognised by DAP-seq.

Thanks the reviewer for the constructive comments. The motif of Fig. 4D in the original manuscript is quite different from the typical LTTR box motif. The motif predicted by MEME-ChIP was based on all the 400 peaks of DAP-seq. In the revised manuscript, all peaks are sorted out and only peaks with a length greater than 500 bp and located in the region -1000 bp to 100 bp relative to the transcription start site are retained for MEME-ChIP analysis. The resulting consensus sequence is AT-N₁₁-AT, which is similar to the motif of LTTR OxyR from *Escherichia coli*. Two peaks containing this consensus binding sequence were random selected for EMSA verification, and the results also showed that AT-N₁₁-AT box is a potential consensus binding sequence of BysR. Detailed results are shown in Line 170-187 “A total of 134 peaks, with a length greater than 500 bp and located in the region -700 bp to 100 bp relative to the transcription start site of the corresponding genes, were used for motif prediction using MEME-ChIP (26) and the conserved motif-like AT-N₁₁-AT box (e-value = 3.8e-021) was found in 95 out of 134 peaks (71%) (Fig. 4D and Supplementary Table S6). This conserved A+T rich box was similar to the LTTR OxyR consensus binding sequence (27) and the T-N₁₁-A motif, typical of LTTRs, was also recognized in this conserved box (28). Thus we proposed that the AT-N₁₁-AT box was the binding feature of BysR (Fig. 4D). Further, the potential BysR binding sequences in 95 peaks were searched by regular expression module in python, and at least one AT-N₁₁-AT box sequence was found in each of these 95 peaks (Supplementary Table S6), implying that BysR binds essentially around the proposed consensus loci to regulate the transcription of the target genes. In order to confirm that the AT-N₁₁-AT box sequences are essential for BysR binding, the promoter region fragments of genes DM992_38905 and DM992_31485, with wild-type and mutated sequences of motif-like consensus sequences, were labeled with Cy5 and used for EMSA analysis. The presence of shift bands in the wild-type probes (probe 4 and probe 6) and the absence of BysR binding to the mutated probes (probe 5 and probe 7) indicate that the predicted consensus-binding sequences are necessary for BysR to bind to the promoter region and regulate the transcription of the targeted genes (Supplementary Fig. S3)” and Fig 4D, Supplementary Table S6, Fig S3 in the revised manuscript.

The sequence of probe3(upstream *ambR1*) used for EMSA analysis overlaps with the peak Merged-Chr3-148813-2 recognised by DAP-seq, and the consensus binding sequence (ATCGGCGATTTTCAT) exists in the overlapping sequence. Please see Supplementary Fig. S4.

3. AmbR1/AmbR2 binding sites, and the molecular mechanism of transcription regulation of these LuxR like regulators are unknown, even if AmbR1 seems to be involved in the expression of many other genes besides *ocf* genes in another Burkholderia sp.(see Ref. 2 in the paper). Authors should investigate the AmbR1 binding sequences present in the surroundings of codon start of *ocfL*, *ambR2* (see Ref 12, Gu et al. AEM, 2011) and perhaps *ocfN* putative promoters by EMSA tests.

We strongly agree with the reviewer's opinion. The study on how AmbR1 regulates *ocf* gene cluster is one of the focus of our future research. As this paper mainly studies the regulatory function of BysR,

so the binding of AmbR1 in the promoter region sequences of *ocfL*, *ambR2*, *ocfN* and other genes will be described in another paper.

4. Because of the absence of the analyses to be done in point 1 and 3, the model proposed in Fig7D is wrong and seems to suggest the presence of a mistake: an arrow suggests the presence of a transcription start site directed toward the translation direction of the *ocf* genes ORFs. In all organisms in traslatino the ORF reading goes from the start to the stop codons found from the 5' to the 3' end of the mRNA and 5' to 3' is also the direction of the transcription. Transcription and translation are codirected, and in bacteria coupled, in particular this aspect is important for transcription/translation regulation of operons.

Many thanks to the reviewer's valuable comments. According to the reviewer's suggestion, we reworked the model. Please see Fig. 7E in the revised manuscript.

5. Independently of the language quality the Discussion is confused and careless and the reasoning badly organised.

According to the reviewer's advice, we have checked up the manuscript carefully and rewrite the discussion.

Kind regards.

A.P zhiguo E

State Key Laboratory of Rice Biology,
China National Rice Research Institute,
Hangzhou, 311400

China

Phone: 86-571-63370586

Email: ezhiguo@caas.cn

January 19, 2023

Dr. Zhiguo E
China National Rice Research Institute
Hangzhou
China

Re: Spectrum02684-22R1 (BysR, a LysR-type Pleiotropic Regulator, Controls Production of Occidiofungin by Activating the LuxR-type transcriptional regulator AmbR1 in Burkholderia sp. JP2-270)

Dear Dr. Zhiguo E:

Thank you for submitting your revised manuscript to Microbiology Spectrum. I have received valuable comments from the third expert based on your revised manuscript. The comments were positive. However, the reviewer and I agree that at this point, significant revisions are still needed for publication. The comments are listed below.

When submitting the revised version of your paper, please provide (1) point-by-point responses to the issues raised by the reviewers, including line numbers of the revised texts, as file type "Response to Reviewers," not in your cover letter, and (2) a PDF file that indicates the changes from the original submission (by highlighting or underlining the changes) as file type "Marked Up Manuscript - For Review Only". Please use this link to submit your revised manuscript - we strongly recommend that you submit your paper within the next 60 days or reach out to me. Detailed instructions on submitting your revised paper are below.

Link Not Available

While we are willing to consider a revised version of this paper at Spectrum, it would be in your best interest to improve the writing. I recommend that you ask a colleague of yours who is a native English speaker to read and provide you some feedback on the writing. You are also welcome to use one of the services here: <https://journals.asm.org/content/language-editing-services>

Sincerely,

Montarop Yamabhai

Journals Department
Reviewer comments:

Reviewer #3 (Public repository details (Required)):

RNASeq data

Reviewer #3 (Comments for the Author):

This manuscript is to report a LysR-type regulator BysR that regulates multiple cellular functions in *Burkholderia* sp. JP2-270, which include production of the antifungal occidiofungin. Routine deletion mutagenesis was conducted. The EMSA assay showed that BysR regulates *ambR1*, a key regulator for the *ocf* gene cluster through direct binding to its promoter region. The study also shows that occidiofungin produced by JP2-270 is the main substance inhibiting *M. oryzae* and BysR controls occidiofungin production by directly activating the expression of *ambR1*. In addition, the transcriptomic analysis revealed altered expression of more than 300 genes in response to *bysR* deletion. Collectively, the authors claim that BysR is a novel pleiotropic regulator in the bacterial strain. The research is original and novel, and most of experiments were conducted with appropriate approaches. Most of the conclusions were drawn with the data support. The following are the major concerns and/or suggestions.

1. The authors show that BysR not only regulates secondary metabolites but also participates in the regulation of various core metabolic pathways, such as the TCA cycle, amino acid synthesis and metabolism, and DNA replication. If so, the BysR mutant should have a very strong phenotype. Have you measured the doubling time of the mutant and growth curve on a rich culture medium and minimum medium?
2. The vector pBBR1MCS-2 was conducted as a cloning vector (Kovach et al. 1995). Why do you think it is an expression vector? Do you have any evidence to show the cloned gene is overexpressed (Table S1; Line 502-510)?
3. Did you test expression level of *recA* (a control for qPCR) in the *bysR* mutant as compared with the wild type? Your data show Δ *bysR* affects amino acid synthesis and metabolism, and DNA replication. It is possible that translation and transcription of *recA* may be affected by the *bysR* mutation.
4. It is hard to follow about the evolutionary analysis. The conclusion is the *ocf* gene cluster is acquired via both horizontal and vertical transfer. Which one is more possible?
5. Presentations of the data and manuscript preparation need to be improved significantly. Its hard to recognize the color codes in Fig. 3D. What does grey, purple blue stand for? Some parts of Fig. 1 could be in supplementary materials. Some supplementary data, such as RNAseq data, can be submitted to NCBI instead to include in supplementary materials. Some sections could be combined and revised. In addition, English proofreading is needed to avoid spelling and grammatical errors.

The following are minor suggestions:

1. Suggest move Table S1 to the main manuscript
2. Suggest moving Fig. 1 to supplementary materials.
3. Naming the mutants should be revised. Δ *bysR* means a genotypic deletion of the *bysR* gene.
4. Lines 218-235: Suggest combining the two paragraphs.
5. Kovach et al. 1995 not found.
6. Line 112: regulates
7. Lines 161-162: incomplete sentence.
8. Line 496: Sources of the organisms used in the study, such as *R. solani* GD118
9. Line 499: specific days and culture media should be indicated and in Figs showing antifungal activities.
10. Line 504: provide the genome accession number.
11. Suggest adding PCR confirmation data of the resulting mutants as supplementary materials (Line 521-523).
12. Lines 525-527: Please provide inoculation quantity or OD values.

Staff Comments:

Preparing Revision Guidelines

Please return the manuscript within 60 days; if you cannot complete the modification within this time period, please contact me. If

you do not wish to modify the manuscript and prefer to submit it to another journal, please notify me of your decision immediately so that the manuscript may be formally withdrawn from consideration by Microbiology Spectrum.

Dear reviewers:

Thank you very much for giving us constructive suggestions on our manuscript entitled “BysR, a LysR-type Pleiotropic Regulator, Controls Production of Occidiofungin by Activating the LuxR-type transcriptional regulator AmbR1 in *Burkholderia* sp. JP2-270” (Spectrum02684-22). These insightful comments and suggestions would help us both in English and in depth to improve the quality of the paper.

We have studied reviewers’ comments carefully. According to the reviewers’ detailed suggestions, we have made a careful revision on the original manuscript. Below is our responses to the reviewers’ comments. The reviewer’s comments are in normal and our responses are in bold with line numbers referring to the revised manuscript.

Reviewer #3 (Comments for the Author):

This manuscript is to report a LysR-type regulator BysR that regulates multiple cellular functions in *Burkholderia* sp. JP2-270, which include production of the antifungal occidiofungin. Routine deletion mutagenesis was conducted. The EMSA assay showed that BysR regulates *ambR1*, a key regulator for the *ocf* gene cluster through direct binding to its promoter region. The study also shows that occidiofungin produced by JP2-270 is the main substance inhibiting *M. oryzae* and BysR controls occidiofungin production by directly activating the expression of *ambR1*. In addition, the transcriptomic analysis revealed altered expression of more than 300 genes in response to *bysR* deletion. Collectively, the authors claim that BysR is a novel pleiotropic regulator in the bacterial strain. The research is original and novel, and most of experiments were conducted with appropriate approaches. Most of the conclusions were drawn with the data support. The following are the major concerns and/or suggestions.

1. The authors show that BysR not only regulates secondary metabolites but also participates in the regulation of various core metabolic pathways, such as the TCA cycle, amino acid synthesis and metabolism, and DNA replication. If so, the BysR mutant should have a very strong phenotype. Have you measured the doubling time of the mutant and growth curve on a rich culture medium and minimum medium?

Many thanks to the reviewer’s valuable comment. We performed the growth curve analysis for WT and Δ *bysR* in rich medium. In LB rich medium, the growth rate of Δ *bysR* was slightly lower than that of wild-type JP2-270, and the biomass of Δ *bysR* was less than JP2-270 in stable period. Both WT and Δ *bysR* showed limited growth in minimal medium, so we did not measure the growth curve of WT and Δ *bysR* cultured in minimal medium. The detailed description were presented in line 168-170 and line 400-405 in the revised manuscript.

2. The vector pBBR1MCS-2 was conducted as a cloning vector (Kovach et al. 1995). Why do you think it is an expression vector? Do you have any evidence to show the cloned gene is overexpressed (Table S1; Line 502-510)?

Many thanks to the reviewer for pointing out this mistake. When constructing the *ambR1* overexpression vector, we actually cloned a constitutive promoter (pCS) into the MCS of the vector pBBR1MCS-2 in advance. The *ambR1* gene coding sequence was cloned downstream of the constitutive promoter. The overexpression of *ambR1* gene was also detected in JP2-270 derived strain containing pBBR2pCS-*ambR1* by qPCR (Figure 6D in the revised manuscript). The modified vector was used for other research, and the description of the modified vector was ignored in the original manuscript, I am deeply sorry.

In the analysis of RNA-seq data, it was found that a cold shock protein gene DM992_01545 (accession number CP029824) was not affected by *bysR* mutation, and showed high expression level in both wild type and *bysR* mutant. Therefore, we selected the promoter region of this gene as the constitutive promoter and modified the cloning vector pBBR1MCS-2. Please see line 407-410 in the revised manuscript.

3. Did you test expression level of *recA* (a control for qPCR) in the *bysR* mutant as compared with the wild type? Your data show

Δ bysR affects amino acid synthesis and metabolism, and DNA replication. It is possible that translation and transcription of *recA* may be affected by the *bysR* mutation.

Thanks for the thoughtful advice. In RNA-seq analysis, we found that the FPKM value of *recA* gene (DM992_06725) in Δ bysR was close to that in the wild type background (RNA-seq raw data: GSE193778), so we selected *recA* gene as a control for qPCR. In qPCR analysis, we also found that the Ct value of *recA* in Δ bysR was basically similar to that in wild type (see raw data for qPCR). So, we think that the transcription of *recA* may not be affected by *bysR* mutation.

4. It is hard to follow about the evolutionary analysis. The conclusion is the *ocf* gene cluster is acquired via both horizontal and vertical transfer. Which one is more possible?

Many thanks for the good advice. We have redescribed the result of evolutionary analysis, please see line 215-224 in the revised manuscript. The *ocf* gene cluster was horizontal transferred between some species, such as *B. contaminans* and *B. vietnamiensis*, *B. stabilis* and *B. anthina*. Meanwhile, the *ocf* gene cluster descended vertically within the genus. Thus, the *ocf* gene cluster was transferred vertically within the genus and horizontally between the species, both horizontal and vertical transfer are the evolutionary processes driving *ocf* gene cluster evolution.

5. Presentations of the data and manuscript preparation need to be improved significantly. Its hard to recognize the color codes in Fig. 3D. What does grey , purple blue stand for? Some parts of Fig. 1 could be in supplementary materials. Some supplementary data, such as RNAseq data, can be submitted to NCBI instead to include in supplementary materials. Some sections could be combined and revised. In addition, English proofreading is needed to avoid spelling and grammatical errors.

We agree the reviewer's good advice. We have moved Fig. 1 to supplementary materials as Supplementary Figure S1 in the revised manuscript.

We also have indicated what the colors mean in Fig. 1 in the revised manuscript, for example, the grey stand for transmembrane protein and purple blue stand for ATPase complex.

We have submitted the RNA-seq and DAP-seq data to NCBI, the accession number are GSE193778 for RNA-seq and GSE193916 for DAP-seq. Meanwhile, in order to facilitate direct reading, we presented these data in supplementary materials.

We have combined the second and third, the seventh and eighth sections of the original results, please see line 105-174 and line 237-267 in the revised manuscript.

We have carefully examined the manuscript for spelling and grammar errors and corrected them.

The following are minor suggestions:

1. Suggest move Table S1 to the main manuscript

According to the reviewer's good advice, we have moved Table S1 to the main manuscript. Please see Table 1 in the revised manuscript.

2. Suggest moving Fig. 1 to supplementary materials.

We agree the reviewer's good advice. We have moved Fig. 1 to supplementary materials. Please see the Supplementary Figure S1 in the revised manuscript.

3. Naming the mutants should be revised. Δ bysR means a genotypic deletion of the *bysR* gene.

Thanks for the advice. Actually, the mutation type of *bysR* is the genotypic deletion in this study. Since the *bysR* mutant was obtained in our previous study (Song *et al.*, 2019), the experimental methods were not included in this manuscript. Please see Table 1 in the revised manuscript and refer Song *et al.*, 2019.

4. Lines 218-235: Suggest combining the two paragraphs.

Thanks for the good advice, we have combined the two paragraphs. Please see line 237-267 in the revised manuscript.

5. Kovach et al. 1995 not found.

Sorry, we missed this reference in the original manuscript. Here we have added it in the revised manuscript, please see line 187 and line 629.

6. Line 112: regulates

Sorry, we have corrected the error, please see line 105 in the revised manuscript.

7. Lines 161-162: incomplete sentence.

Many thanks to the reviewer for pointing out this mistake. We have combined the second and third sections of the original results, and removed the incomplete sentence. Please see line 105-174 in the revised manuscript.

8. Line 496: Sources of the organisms used in the study, such as *R. solani* GD118

According to the good advice, we have added the sources of the organisms. Please see Table 1 in the revised manuscript.

9. Line 499: specific days and culture media should be indicated and in Figs showing antifungal activities.

According to the reviewer's good advice, we have added the specific days and culture media in Line xxx and in Figs showing antifungal activities. Please see line 394-397, Fig. 2 and Fig. 4 in the revised manuscript.

10. Line 504: provide the genome accession number.

According to the good advice, we have added the genome accession number, please see line 412 in the revised manuscript.

11. Suggest adding PCR confirmation data of the resulting mutants as supplementary materials (Line 521-523).

Thanks for the thoughtful advice. We have added the PCR confirmation data of the resulting mutant in the supplementary materials, please see Supplementary Figure S6.

12. Lines 525-527: Please provide inoculation quantity or OD values.

We agree the reviewer's good advice. We have added the inoculation quantity in the revised manuscript. Please see line 436-437 in the revised manuscript.

Kind regards.

A.P zhiguo E

State Key Laboratory of Rice Biology,

China National Rice Research Institute,

Hangzhou, 311400

China

Phone: 86-571-63370586

Email: ezhiguo@caas.cn

February 24, 2023

Dr. Zhiguo E
China National Rice Research Institute
Hangzhou
China

Re: Spectrum02684-22R2 (BysR, a LysR-type Pleiotropic Regulator, Controls Production of Occidiofungin by Activating the LuxR-type transcriptional regulator AmbR1 in Burkholderia sp. JP2-270)

Dear Dr. Zhiguo E:

Thank you for your revision. Congratulations, your manuscript has been accepted, and I am forwarding it to the ASM Journals Department for publication. You will be notified when your proofs are ready to be viewed.

Sincerely,

Montarop Yamabhai
Editor, Microbiology Spectrum
